# Transmission of antimicrobial resistance in the gut microbiome of gregarious cockroaches: the importance of interaction between antibiotic exposed and non-exposed populations

Amalia Bogri,[1] Emilie Egholm Bruun Jensen,[1] Asbjørn Vedel Borchert,[1] Christian Brinch,[1] Saria Otani,[1] Frank M. Aarestrup[1]

**ABSTRACT** Antimicrobial resistance (AMR) is a major global health concern, further complicated by its spread via the microbiome bacterial members. While mathematical models discuss AMR transmission through the symbiotic microbiome, experimental studies are scarce. Herein, we used a gregarious cockroach, *Pycnoscelus surinamensis,* as an *in vivo* animal model for AMR transmission investigations. We explored whether the effect of antimicrobial treatment is detectable with metagenomic sequencing, and whether AMR genes can be spread and established in unchallenged (not treated with antibiotics) individuals following contact with treated donors, and under various frequencies of interaction. Gut and soil substrate microbiomes were investigated by metagenomic sequencing for bacterial community composition and resistome profiling. We found that tetracycline treatment altered the treated gut microbiome by decreasing bacterial diversity and increasing the abundance of tetracycline resistance genes. Untreated cockroaches that interacted with treated donors also had elevated tetracycline resistance. The levels of resistance differed depending on the magnitude and frequency of donor transfer. Additionally, treated donors showed signs of microbiome recovery due to their interaction with the untreated ones. Similar patterns were also recorded in the soil substrate microbiomes. Our results shed light on how interacting microbiomes facilitate AMR gene transmission to previously unchallenged hosts, a dynamic influenced by the interaction frequencies, using an *in vivo* model to validate theoretical AMR transmission models.

**IMPORTANCE** Antimicrobial resistance is a rising threat to human and animal health. The spread of resistance through the transmission of the symbiotic gut microbiome is of concern and has been explored in theoretical modeling studies. In this study, we employ gregarious insect populations to examine the emergence and transmission of antimicrobial resistance *in vivo* and validate modeling hypotheses. We find that antimicrobial treatment increases the levels of resistance in treated populations. Most importantly, we show that resistance increased in untreated populations after interacting with the treated ones. The level of resistance transmission was affected by the magnitude and frequency of population mixing. Our results highlight the importance of microbial transmission in the spread of antimicrobial resistance.

**KEYWORDS** cockroaches, metagenomics, antimicrobial resistance, antibiotics, tetracycline, microbiome, bacterial transmission

Antimicrobial resistance (AMR) is an increasing global health crisis, threatening to diminish the efficacy of antibiotics, which have long served as a cornerstone for modern medicine (1, 2). While the direct use of antibiotics has been frequently implicated in the rise of AMR, the role of the non-pathogenic (symbiotic) microbiome bacteria

Address correspondence to Amalia Bogri, amabog@food.dtu.dk.

The authors declare no conflict of interest.

See the funding table on p. 19.

in the propagation and dissemination of resistance genes, and how they transmit within complex communities, is increasingly recognized as a key facilitator of this phenomenon (3–6). Bacteria in the microbiome can act as reservoirs for antimicrobial resistance genes (ARGs), and the potential for horizontal gene transfer between these and pathogenic bacteria can lead to the emergence of multidrug-resistant pathogens, exacerbating the AMR problem (7).

While mathematical models have been valuable tools for predicting the potential trajectories and outcomes of AMR selection and dissemination, there remain major gaps in our understanding when it comes to actual, real-world transmission events (8). Experimental investigations into AMR transmission via microbiomes across hosts have been scant, not least because of the inherent challenges in setting up such studies (8, 9). There is, however, a pressing need for tractable, *in vivo* models to test hypotheses and validate mathematical predictions. While there have been a number of observational studies (e.g., see references 10–12), up to now there are only very few *in vivo* experimental studies (e.g., see references 13, 14) on the effect of antimicrobial treatment on the microbiome. The limited existing experimental studies on AMR transmission dynamics within live microbiomes reveal shifts in AMR gene acquisitions resulting from host-host interactions but lack detailed experimental setup and in-depth analyses.

Insects, particularly those with aggregating behaviors, present unique opportunities for understanding between-host transmission dynamics (15). *Pycnoscelus surinamensis* is a gregarious species known to live in closely knit, clonal colonies (16), offering a potential platform to study such interactions. Moreover, the cockroach gut microbiomes, akin to other insects, are rich in bacterial taxa and play pivotal roles in digestion, nutrient assimilation, and overall health (17). Yet, only a handful of studies have examined the gut microbiome of *P. surinamensis*, while no study has ever explored its resistome (18–21).

With the scarcity of research into AMR transmission within symbiotic microbiomes across hosts, there is an urgent demand for practical, *in vivo* models to rigorously examine and substantiate theoretical predictions (8, 15). Here, we investigated a species of gregarious cockroaches, *P. surinamensis*, as an *in vivo* experimental model for studying AMR transmission. Leveraging metagenomic sequencing, we studied the impact of tetracycline treatment on the gut microbiome over time, where we demonstrate a major increase in tetracycline ARGs. We also examine the effect of mixing treated and untreated populations, and we show that there is transmission of tetracycline ARGs to untreated individuals. We specifically find that a single large mixing event leads to higher ARG levels compared to multiple smaller events. Our study shows the potential of using gregarious cockroaches as an *in vivo* experimental microbiome for research on the transmission and selection of AMR in host populations.

## MATERIALS AND METHODS

### *P. surinamensis* as a model host

*P. surinamensis* has a parthenogenetic lifecycle where populations are primarily, if not exclusively, female and reproduce without mating; hence, *P. surinamensis* colonies are clonal (22), making them a good model to study microbiome changes excluding host variation effects (23). The reproduction is ovoviviparous, with the female gestating the ootheca internally until the hatching of the nymphs (24, 25). As hemimetabolous insects, *P. surinamensis* nymphs undergo several molting stages (instars) for 2–5 months until reaching adulthood (24, 26, 27). The first instars in cockroaches usually shed their gut lining when they molt, so they have to re-establish their microbiome with horizontal transmission (28). The later instars do not completely shed their gut lining when they molt, and they keep a more stable microbiome, similar to the adult's (29). For this experiment, we included and sampled only late-stage instars and supplemented with adults when more quantity was needed.

## Laboratory rearing of *P. surinamensis* colony

The cockroach colony was acquired in 2021 from a commercial insect breeder (blattaria.fr) and kept in a temperature and humidity-controlled room, at 27°C and 50% relative humidity. The colony is housed in plastic terraria (dimensions: 19 × 19 × 19 cm), whose lids are fitted with fine metal mesh for ventilation (mesh size: 1.6 mm). For substrate, we use soil collected from the DTU campus at Kongens Lyngby, Denmark. The cockroaches are fed twice a week with fruits (pear) and provided water as crystals to prevent drowning. Rearing containers are cleaned from food remnants to prevent mold growth. The population's health and behavior are monitored for any anomalies, such as signs of disease, pests, or stress. At the time of our experiment, the colony had been maintained for a year in these conditions.

## Experimental setup

The experiment was conducted for 16 days (Fig. 1). Two hundred and fifty cockroaches were collected and isolated from the main colony (day 0) and starved for 1 day (day 1). On day 1, cockroaches and soil were sampled from the initial population (container A) for metagenomic sequencing. The remaining cockroaches were divided into two containers: A and B. From day 1 to day 7, the cockroaches in terrarium B were treated daily with tetracycline (200 µg per cockroach), whereas cockroaches in terrarium A remain untreated. All cockroaches were fed daily from day 1 with their routine diet (pear). On day 8, cockroaches and soil were sampled from both terraria for metagenomic sequencing. The remaining cockroaches were color-marked according to their treatment status to discriminate between the individuals when they are mixed during the transmission experiment as the following:

Treated and untreated individuals were mixed by transferring tetracycline-treated cockroaches (terraria B) into the untreated population (terraria A). In the mixed population terrarium (A1), equal numbers of treated (donors) and untreated (recipients) individuals were maintained in order not to introduce bias between the populations because of varying population sizes. The remaining treated cockroaches were kept in their terrarium (B1). From day 8 to day 14, the two terraria (A1, B1) were fed daily without any antimicrobial treatment, and cockroach samples were taken on days 10, 12, and 15. From the terrarium with the mixed population (A1), we sampled both treated and untreated individuals separately, as they were marked with different colors. Soil samples from both terraria were taken only on the 15th day.

A parallel experiment was conducted from day 8 to day 15 where the antibiotic stressor was applied more frequently to increase its effect. On day 8, we transferred treated cockroaches and soil into a separate terrarium (B2) and treated them daily with tetracycline until the end of the experiment (days 8–15). We also transferred untreated cockroaches (recipients) and soil in a fourth terrarium (A2), supplemented by treated donors. This population (in A2) received small numbers of treated cockroaches from terrarium B2 every 2 days, i.e., on the days 8, 10, 12, and 15. Cockroach and soil samples were taken on the 15th day from both terraria in this setup for metagenomic sequencing.

## Soil substrate preparation

Soil for the experimental terraria was collected at the DTU campus in Kongens Lyngby, Denmark. To eliminate live viable contamination, the soil was frozen at −80°C overnight and thawed the following day, twice. The soil was subsequently sieved to remove rocks and plant debris. The prepared soil was then distributed to the terraria of the original cockroach colony 1 week before the experiment to avoid additional stress on the cockroaches. As *P. surinamensis* are soil-burrowing cockroaches, approximately 1 cm of soil depth was used for each experimental terrarium.

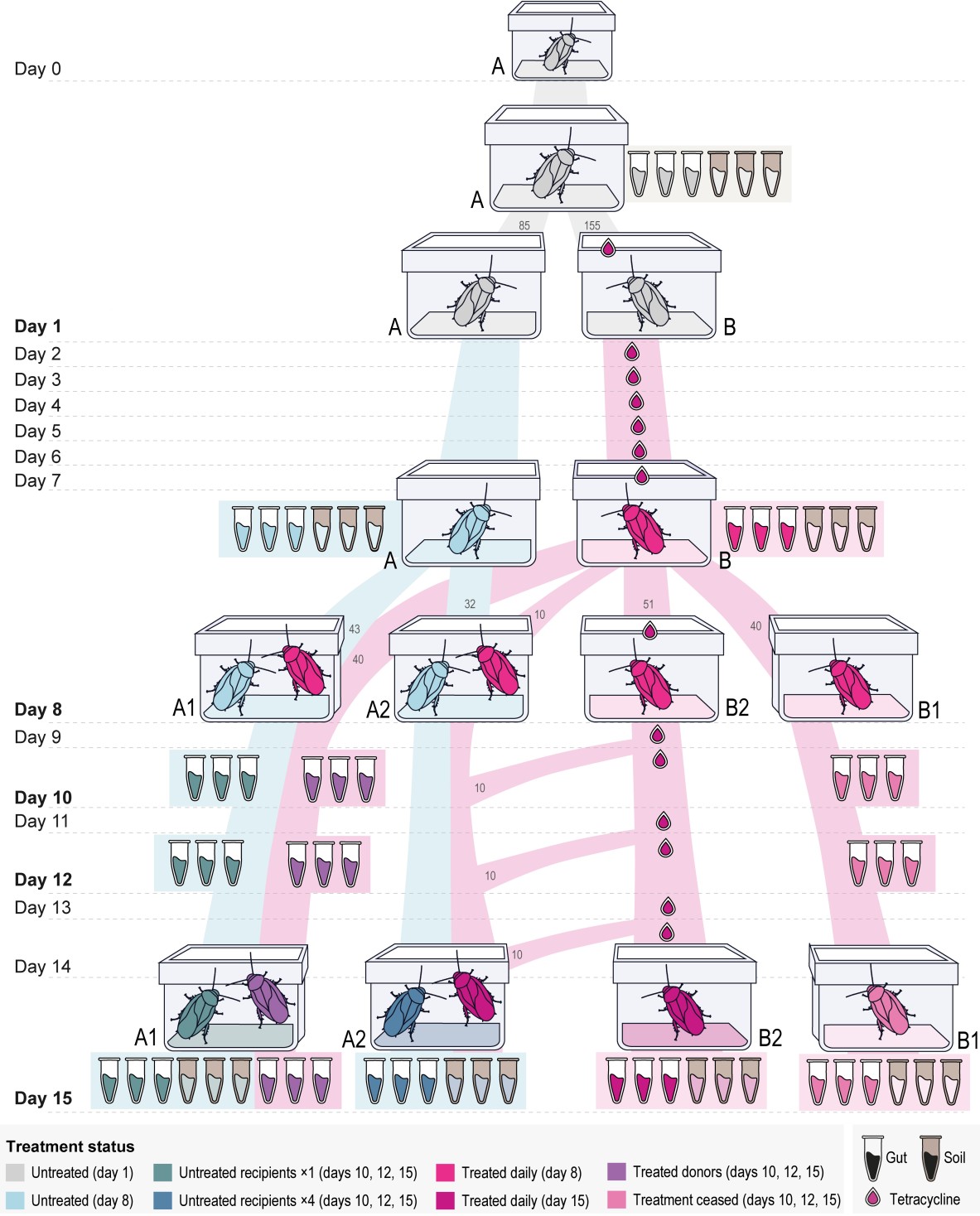

**FIG 1** Experimental set-up of the study from day 0 to day 15. Sampling points are indicated with tubes (days 1, 8, 10, 12, and 15). The destination and number of transferred individuals are illustrated (days 1, 8, 10, 12, and 14). The colors indicate the treatment status of the cockroaches. Symbols: white tube, three pooled guts samples (cockroach); brown tubes, 0.2 g of soil samples; droplets, treatment with 200 µg of tetracycline hydrochloride per cockroach.

## Paint marking of the cockroaches and dietary setup

On the day of population transfer (day 8), the cockroaches were briefly immobilized at low temperature (approximately 5 min at 5°C). A dot was drawn on the pronotum of each cockroach (30) with non-toxic water-based POSCA markers, often used to mark

bees (e.g., see reference 31). The treated cockroaches were marked green, and untreated cockroaches were marked white. Only cockroaches with visible markings were selected for sampling to avoid individuals that molted during the experiment.

The cockroaches were fed daily with 1–2 mm thick slices of pear (washed and refrigerated). The food was monitored and adjusted based on the number of cockroaches per terrarium to avoid dietary leftovers.

## Tetracycline preparation and treatment

Tetracycline hydrochloride (Sigma Aldrich, PHR1041-500MG) was dissolved according to the manufacturer's instructions in 60% ethanol at a concentration of 0.04 g/mL. The dose was set to 200 µg of tetracycline hydrochloride per cockroach. We established in a pilot experiment that such a high antimicrobial concentration does not cause excess mortality in *P. surinamensis*. Tetracyclines have been found to inhibit not only bacteria but also the mitochondria in eukaryotic cells; specifically, in *Drosophila melanogaster* and *Blattella germanica*, tetracyclines delayed and reduced growth and fecundity, and increased mobility early in life (15, 32). The duration of our pilot and our main experiments was short (2 weeks). Thus, as we sampled late-stage juveniles that did not molt during the experiment, we are confident that delays in growth and fecundity would not affect our results. Tetracycline has the potential to affect the mobility of flies by improving muscle fitness, but this effect was only minor later in life (32). We did not perform mobility assays on *P. surinamensis*, and, to our knowledge, tetracycline's effect on mobility has not been explored in hemimetabolous insects. Thus, we sampled only late-stage juveniles, expecting that their mobility is less or not affected by tetracycline. Aliquots of the tetracycline hydrochloride solution were stored at −20°C until used. During the experiment, a prepared aliquot was thawed on ice and vortexed. The amount of tetracycline hydrochloride was calculated and adjusted depending on the number of estimated cockroaches (5 µL per cockroach) in the terrarium. The solution was carefully pipetted onto the pear slices of the terrarium that was receiving treatment (B or B2), and the dietary consumption was visually inspected.

## Sampling and dissection

Nine cockroaches were collected at each sampling point (Fig. 1), each replicate contains three pooled guts, resulting in three replicates of each sampling point. The cockroaches were sampled on days 1, 8, 10, 12, and 15. In total, we have 14 sampling points, and 42 samples comprising 126 guts. The sampled cockroaches were placed in sterile falcon tubes and refrigerated briefly before dissection with sterile forceps for each dissection. The gut was dissected from the abdomen of each cockroach without its surrounding tissues. Three guts were pooled in 500 µL phosphate-buffered saline (PBS) in 1.5 mL LoBind Eppendorf tubes and stored in −20°C until the DNA extraction. As *P. surinamensis* is coprophagous, substrate samples were also collected to investigate if the effect of tetracycline treatment would be traceable in their soil substrate (which contains the cockroach feces). We sampled substrate from the terraria on days 1, 8, and 15, with 7 sampling points and 21 samples. A small amount of soil substrate, ~5 g, from each terrarium was saved in 15 mL falcon tubes at each soil sampling point and stored in the freezer until the DNA extraction.

## DNA purification and sequencing

DNA was extracted from the gut samples with the QIAamp Microbiome Kit (Qiagen, Cat. No.: 51704) that depletes eukaryotic DNA to decrease the cockroach host DNA following the manufacturer's instructions with the following modifications: step 8—incubation time was increased to 40 min; step 10—incubation time was increased to 15 min; step 17—the elution buffer AVE was preheated to 56°C; elution steps 17 and 18 were repeated twice. DNA extraction from soil samples was performed with the DNeasy PowerSoil Pro Kit (Qiagen, Cat. No.: 47014). Approximately 0.2 g of soil was

used following the manufacturer's instructions with the following modifications: step 2 samples were homogenized with TissueLyser II; step 18—he elution buffer Solution C6 was preheated to 56°C; elution steps 18 and 19 were repeated twice. DNA concentration was measured with the Qubit 4 Fluorometer (Invitrogen, Cat. No.: Q33238) and the Qubit dsDNA BR Assay Kit (Invitrogen, Cat. No.: Q33265). Two hundred nanograms of DNA in each sample was used for metagenomic sequencing. The metagenomic sequencing was carried out on Illumina NovaSeq 6000 sequencing platform, and library preparation was performed using KAPA PCR-free kits with minimal PCR cycles (four cycles) on all samples. One sample (day 10, replicate 3 of the ceased-treatment gut samples) failed the library preparation and sequencing.

## Bioinformatics and quality control

We performed all bioinformatics analysis on the Danish National Life science supercomputer, Computerome2. Quality control and trimming were performed on the raw reads with an in-house pipeline, FastQC v. 0.11.5, utilizing bbduk2 from BBTools suite v. 36.49 of NGS tools (Bushnell, BBMap). Adapters were identified and removed using 19-mers. Right-end trim was applied to bases with Phred scores below Q20, ensuring a base call accuracy of at least 99%. Reads ending up being shorter than 50 bp were discarded. General data analysis was carried out in Python and R.

### Taxonomic assignment of bacterial community

The trimmed reads were aligned and mapped with KMA v. 1.4.2 (33) against a custom reference genomic database (last updated 24.05.2022 [34]). The custom database comprised NCBI GenBank databases of bacteria (closed genomes), archaea, MetaHitAssembly (PRJEB674–PRJEB1046), HumanMicrobiome (genome assemblies), bacteria_draft, plasmid, virus, fungi, protozoa, and parasites; taxonomic assignment was carried out as in reference 34. A total of ~174 million sequence fragments, from 62 samples, were assigned taxonomically, with a median of ~2 million fragments per sample. From each output mapstat file, the number of aligned fragments was used to create a taxon abundance table for all samples, to be used for the downstream analysis. The number of aligned fragments for each taxon was corrected according to their reference length. For this, we estimated the ratio of the references' genome size over the median genome size of the reference's superkingdom within the genomic reference file. We then divided the number of fragments mapped to each reference, for each sample, by the estimated ratio. We produced abundance tables for each taxonomic level by summing the fragment counts of lower taxonomic levels, discarding fragments assigned to "unknown." For the downstream diversity analyses, only bacterial reads were included. For the bacterial community analyses, we present the data at the taxonomic levels of phylum and genus. We removed features that had a sum of less than 100 mapped reads in the entire data set.

### Antimicrobial resistance quantification

Similar to the above, the trimmed reads were aligned with KMA v. 1.4.2 (33) to the ResFinder database (v. 20200125 [35, 36]) of known and acquired resistance genes, to create ARG abundance tables for all the samples. A total of ~70,000 sequence fragments were assigned taxonomically, with a median of ~600 fragments per sample. The number of aligned fragments of each gene was adjusted for their ResFinder reference template length, by dividing by the length of the reference gene (in kilobases). We then binned variants of ARGs to close homolog groups of 90% identity, as in reference 37, and we used the representative sequences to name the groups. We also binned the ARG variants to the level of drug class, as in reference 37. To quantify tetracycline ARGs in each sample, we obtained the relative abundance of fragments per kilobase reference per million bacterial fragments (FPKM). For the rest of the data analyses, we removed features that had a sum of less than 100 mapped reads in the entire data set.

To examine the context of the identified ARGs, we employed ARGextender on the trimmed reads, which recursively applies KMA v. 1.4.2 (33) and SPAdes 3.15.5 (38, 39): KMA identifies target ARG sequences in each sample, and then SPAdes carries out *de novo* assemblies of the reads matching each ARG target, as in Martiny et al. (submitted for publication). Then, we explored the flanking regions of the extended ARGs with Flankophile (40) (https://bitbucket.org/genomicepidemiology/flankophile), a bioinformatic pipeline for flanking region analysis.

## Data analysis

The α-diversity of the bacterial community of each sample was estimated at the taxonomic level of genus with the exponential Shannon index (effective number of genera), which takes into account both the richness and the evenness of the community (41). We visualized both the bacterial and the ARG composition of each sample with barcharts at phylum and genus level, and AMR class and gene level. The β-diversity of the gut microbiomes and ARG content was explored compositionally (42–44) with ordination analysis; we used principal component analysis (PCA) (Fig. S1) on CLR transformed data, as CLR coefficients obey Euclidean geometry (44, 45). For this, the features were filtered in order to achieve a lower number of features than samples. We kept features with a high CLR median to avoid features with low abundance, and features with a high CLR variance, to exclude features that are not variant between samples. After filtering the features of the fragment count data set, Bayesian zero replacement and CLR transformation were performed for the PCA using the pyCoDaMath package (https://bitbucket.org/genomicepidemiology/pycodamath) in Python. The filtered data sets were also analyzed with the ALDEx2 package in R (46–48) to determine the statistically significant (wi.eBH <0.05) and differentially abundant features (|effect| > 1) between sample groups.

## RESULTS

### Emergence and transmission of ARGs in *P. surinamensis* gut microbiome

#### Baseline AMR before treatment

To characterize the baseline AMR of the natural gut microbiome of *P. surinamensis*, we examined the untreated resistome of day 1 and day 8 of the experiment. The overall ARG content, and especially the tetracycline ARG content, was extremely low and mostly conferred resistance to aminoglycosides, tetracycline, and beta-lactams (Fig. 2A). The most abundant ARGs were *aadA*11, *oqx*B, and *aac*(6')-*lc* (Fig. 2A; Fig. S2A). Specifically for ARGs that conferred tetracycline resistance, we recorded a low abundance of *tet*(S/M), *tet*(M), and *tet*(O) (Fig. 2A; Fig. S2A).

#### Tetracycline ARGs increased after treatment

Tetracycline treatment led to an increase of tetracycline ARG levels, with tetracycline ARGs taking up more than 60% of the total ARG composition (Fig. 2A). Specifically, the genes *tet*(S/M) and *tet*(M) were significantly increased in the daily-treated microbiomes, and the genes *tet*(O) and *tet*(Q) were also differentially abundant with a smaller effect (Fig. 3A). This was further confirmed by the ordination analysis, where all four genes clearly drove the difference between the treated and the untreated samples (Fig. 3A). ARGs that confer resistance to other antimicrobial classes, namely, *aadA*11, *oqx*B, *qnrE*1, and *aac*(6')-*lc*, present in the untreated samples were also found in the daily-treated samples (Fig. S2A).

#### Tetracycline ARGs were transmitted to untreated populations

The transfer of treated individuals (B) into the untreated populations (A1, A2) increased the tetracycline ARG levels of the recipients. The increase was clear already 2 days after transfer, on day 10, and remained stable until the end of the experiment on day 15 (Fig.

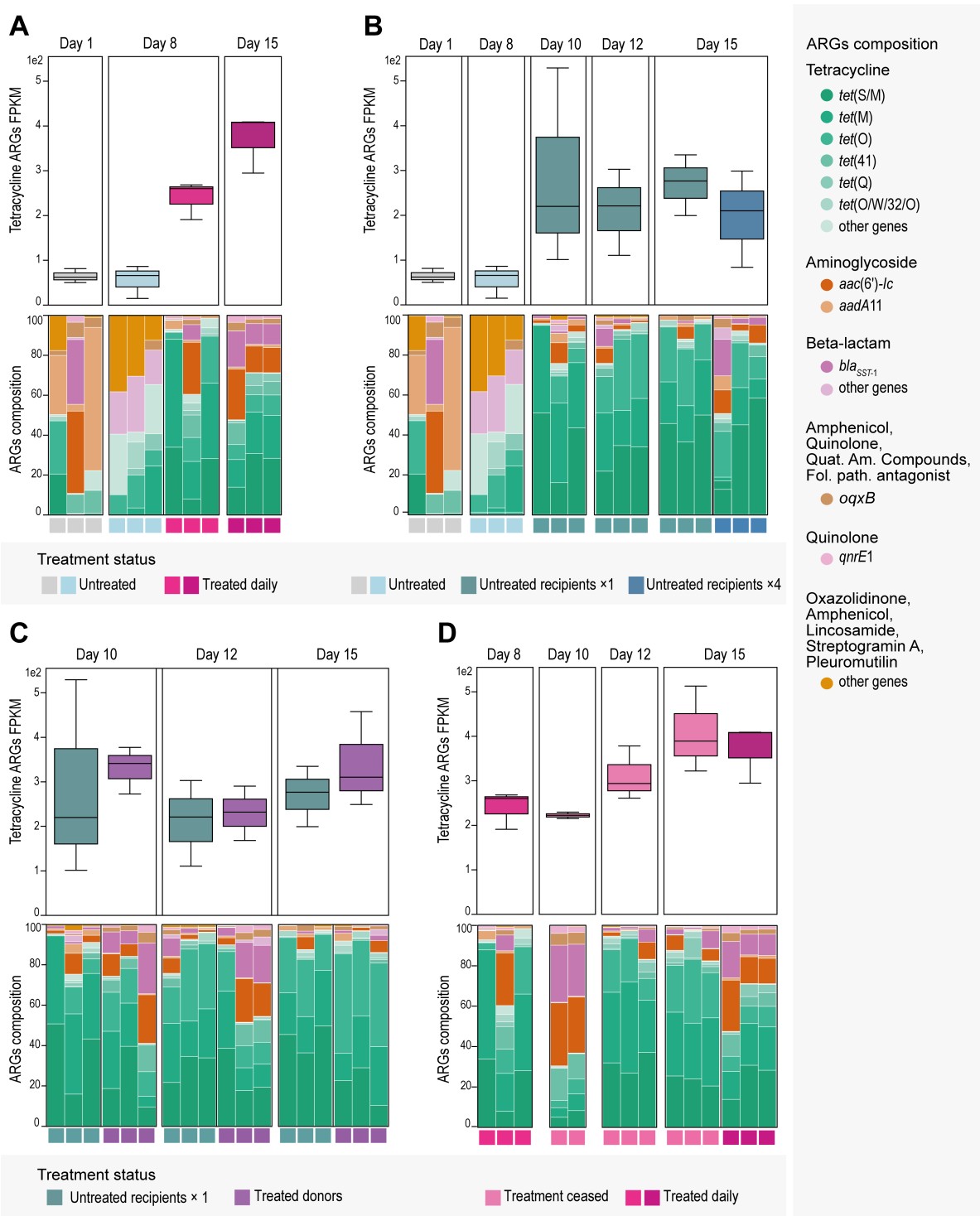

**FIG 2** ARG dynamics in the gut microbiome samples ($n$ = 41). Box plots: tetracycline ARGs relative abundance (FPKM). Box colors indicate treatment status. Bar plots: ARG composition of the 11 most abundant ARGs across all gut samples. ARGs with low abundance are aggregated in the "other genes" categories based on the antimicrobial class they confer resistance to. Bar colors indicate ARGs. All triplicates are ordered by day and treatment status. (A) Comparison between untreated and daily treated microbiomes. (B) Comparison between untreated microbiomes and untreated recipients. The untreated recipients were mixed with treated individuals in one large event, or four smaller events. (C) Comparison between untreated recipients and treated donors, of the one large mixing event. (D) Comparison between daily-treated and post-treatment microbiomes.

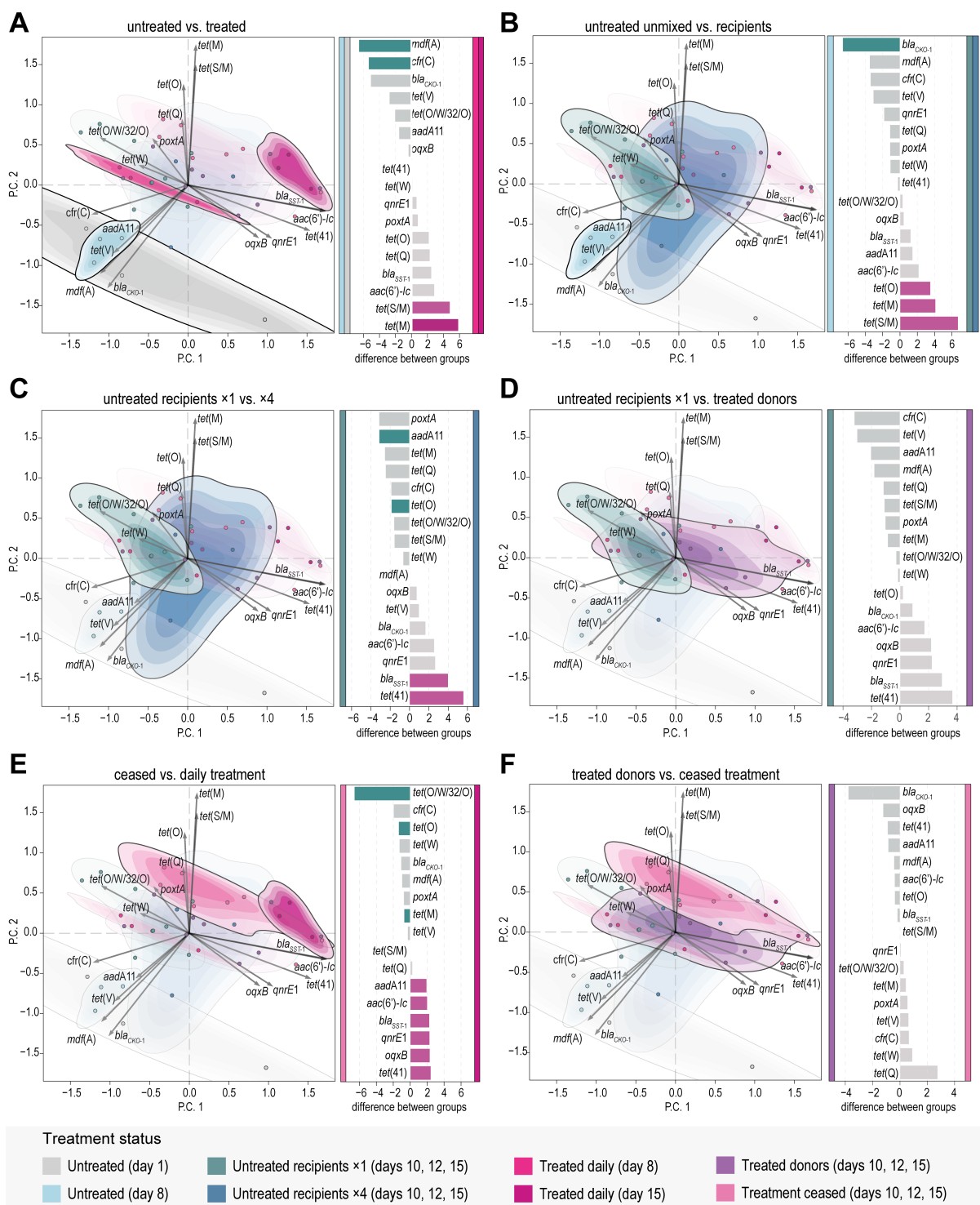

**FIG 3** Differences in ARG composition between the gut microbiome samples ($n = 41$). Ordination plots: principal component analysis on CLR-transformed ARGs fragment counts: P.C. 1 explains 38% of the variation, and P.C. 2 explains 30% of the variation. The same plot is exhibited for panels A–F, with highlights (see Fig. S1A for the non-highlighted version). Point and contour colors indicate treatment status. Barcharts: differential abundance of ARGs between pairs of groups. Gray bars indicate |effect| < 1, and lighter bars indicate wi.eBH > 0.05. (A) Comparison between untreated (days 1 and 8, $n = 6$) and daily-treated (days 8 and 15, $n = 6$) microbiomes. (B) Comparison between untreated (unmixed) (day 8, $n = 3$) and untreated recipient (day 15, $n = 6$) microbiomes. (C) Comparison between untreated recipients of one large mixing event (day 15, $n = 3$) vs of four small mixing events (day 15, $n = 3$). (D) Comparison between untreated recipients (days 10, 12, and 15, $n = 9$) and treated donors (days 10, 12, and 15, $n = 9$) of the one large mixing event. (E) Comparison between post-treatment (day 15, $n = 3$) and daily-treated microbiomes (day 15, $n = 3$). (F) Comparison between treated donor (days 10, 12, and 15, $n = 9$) and post-treatment microbiomes (days 10, 12, and 15, $n = 8$).

2B). The ARG composition of the untreated recipients was distinct from the untreated unmixed microbiomes in the ordination analysis (Fig. 3B). In the majority of the untreated recipients, more than 80% of the AMR content was tetracycline ARGs, with most abundant the *tet*(S/M), *tet*(M), and *tet*(O) ARGs (Fig. 2B; Fig. S2A). These three genes were also found as statistically more abundant in the untreated recipients compared to the untreated unmixed microbiomes (Fig. 3B).

Tetracycline ARG levels in untreated recipients who received a single large amount of treated donors (on day 8) were slightly higher than those in untreated recipients who received multiple smaller amounts of treated donors (four times on days 8, 10, 12, and 14) at the end of the experiment (Fig. 2B). This is also visible in the ordination analysis, where the one-transfer recipients are closer to all treated samples, whereas the four-transfer recipients cluster closer to the unmixed untreated ones (Fig. 3C). The one-transfer recipients had higher levels of *aadA*11 and *tet*(O), whereas the four-transfer recipients had higher levels of $bla_{SST-1}$ and *tet* (42) (Fig. 3C) even though all four ARGs were in low relative abundance (Fig. S2A).

The tetracycline ARG levels in untreated recipient samples were high and similar to those in treated donors (Fig. 2C). The ARG composition was also similar between recipients and donors, as shown in the ordination analysis (Fig. 3D). No significant difference in ARG abundances was found between the two groups (Fig. 3D). This is further supported by the flanking region analysis of the ARGs, which indicated that it were the same *tet*(S/M) and *tet*(M) genes found in untreated recipients and treated donors (Fig. S5A and B).

### Tetracycline ARGs increased post treatment

The level of tetracycline AMR kept increasing even after the treatment was stopped. By the end of the experiment, the daily-treated samples had the same tetracycline AMR level as the samples that stopped receiving treatment and were left to recover for 8 days (Fig. 2D). Post-treatment samples generally did not overlap with the daily-treated samples in the ordination analysis, suggesting different ARG compositions (Fig. 3E). Indeed, by day 15, post-treatment samples had differentially abundant genes like *tet*(O/W/32/O), *tet*(O), and *tet*(M), whereas the daily-treated samples were differentially abundant with *tet* (42) and other non-tetracycline ARGs (*oqx*B, *qnrE*1, $bla_{SST-1}$, *aadA*11, and *aac*(6′)-*Ic*) (Fig. 3E).

The tetracycline AMR level of the treated donors also increased during the 8 last days of the experiment, wherein they did not receive treatment (Fig. 2D). Their tetracycline ARG increase was similar to post-treatment samples, only slightly less (Fig. 2C and D); with no significantly different ARG abundances identified, as expected by their similarity in the ordination analysis (Fig. 3F).

## Bacterial community dynamics of *P. surinamensis* gut microbiome

### Natural gut microbiome before treatment

To characterize the natural bacterial microbiome of *P. surinamensis*, we analyzed the 6 untreated microbiomes from the 1st and 8th day of the experiment (Fig. S3A). The most abundant phyla were Firmicutes (Bacillota), Proteobacteria (Pseudomonadota), Actinobacteria (Actinomycetota), Bacteroidetes (Bacteroidota), and Verrucomicrobia (Verrucomicrobiota) (Fig. 4A). The untreated microbiomes from the 1st and 8th day had the highest bacterial diversity at the genus level, accounting for richness and evenness (Fig. 4A). Within Firmicutes, the most common genera belonged to Bacilli, namely, *Enterococcus*, *Lactococcus*, *Paucilactobacillus*, *Loigolactobacillus*, and *Lacticaseibacillus* (Fig. S3A). In Proteobacteria, the genera with the highest relative abundance belonged to Gammaproteobacteria, with *Pseudocitrobacter* and *Azomonas*, and to Alphaproteobacteria with *Devosia*, *Mesorhizobium*, and *Ensifer* (Fig. S3A). *Serratia* had a high relative abundance in the 1st day samples (Fig. S3A). Within Actinobacteria, the genus *Mycolicibacterium* had a high relative abundance, particularly on the 8th day samples, which were fed daily (Fig. S3A). In the same samples, *Blattabacterium* of the

phylum Bacteroidetes was also prominent (Fig. S3A). Finally, one member of the phylum Verrucomicrobia, *Ereboglobus*, had a high relative abundance on the 1st-day samples (Fig. S3A).

## Treatment enriched Firmicutes and decreased diversity

We examined the effect of tetracycline treatment by analyzing the microbiomes that were treated daily for 8 and 15 days. Microbiome bacterial diversity more than halved after treatment based on the exponential Shannon index (Fig. 4A). This diversity decline can be attributed to a decrease in the evenness of the bacterial community (Fig. S4A). The treatment had a clear effect on the bacterial composition, with a notable enrichment of Firmicutes, which surpassed 70% of the phyla composition (Fig. 4A). This is attributed to the increase of *Lactococcus* and *Solibacillus*, which were differentially abundant between the untreated and the treated microbiomes (Fig. 5A; Fig. S3A). There was also a clear decrease of Proteobacteria, with a significant decrease of *Pseudocitrobacter* and *Desulfovibrio* (Fig. 4A). Yet, Proteobacteria remained present, with a significant increase in *Serratia* and *Nocardioides* (Proteobacteria) (Fig. 4A). These effects were also visible in the ordination analysis, where the treated and untreated samples formed distinct clusters (Fig. 4A).

## Microbiome of untreated recipients became similar to treated donors

The transfer of treated individuals into the untreated populations affected the gut microbiome of both donors and recipients. The untreated recipient microbiomes were dominated by Firmicutes, particularly *Lactococcus*, *Enterococcus*, and *Loigolactobacillus*, which took up more than 80% of the composition in the majority of the samples (Fig. 4B; Fig. S3A).

The bacterial diversity of the untreated recipients dropped immediately after the transfer, as measured on day 10, and remained low until the end of the experiment, on day 15 (Fig. 4B), due to the decrease in bacterial evenness (Fig. S4A). The bacterial composition was distinct from the unmixed untreated microbiomes, as shown in the ordination analysis (Fig. 5B). Specifically, in the untreated recipients, there was a significant increase in *Lactococcus*, *Serratia, Luteimonas, Porphyromonas,* and *Secundilactobacillus*, and a decrease in *Citrobacter*, *Geomicrobium*, *Gordonia,* and *Desulfovibrio* when compared to the unmixed untreated microbiomes (Fig. 5B).

The microbiomes of the untreated recipients that received a single large amount of donors, on day 8, were slightly different than the microbiomes of the untreated recipients that received multiple smaller amounts of donors (four times—days 8, 10, 12, and 14). The one-transfer recipients exhibited a slightly lower bacterial diversity compared to the four-transfer recipients (Fig. 4B). The samples of the latter also appear more diluted in the ordination analysis, suggesting variations in community composition within the four-transfer recipients group (Fig. 5C). Despite the small differences, the one-transfer and four-transfer recipients remain quite similar, with only *Serratia* and *Nocardioides* as differentially abundant in the latter (Fig. 5C).

The bacterial diversities of both untreated recipient groups and the treated donors were at similarly low levels (Fig. 4C). Their microbial composition was also similar, with the two groups closely overlapping in the ordination analysis (Fig. 5D). Indeed, there are only a few differentially abundant taxa separating the recipients and the donors: *Desulfovibrio* was more abundant in the recipients compared to the donors, which had significantly more *Serratia* and *Dysgonomonas* in their treated microbiomes (Fig. 5D).

## Microbial diversity of treated donors increased

The bacterial diversity of the treated donors kept increasing after their transfer, as evident on days 10 and 15 (Fig. 4C); yet, it never returned to pre-treatment levels. The treated donors clearly separated in the ordination analysis from the daily-treated microbiomes that were not transferred (Fig. 5E and F), indicating a different microbial

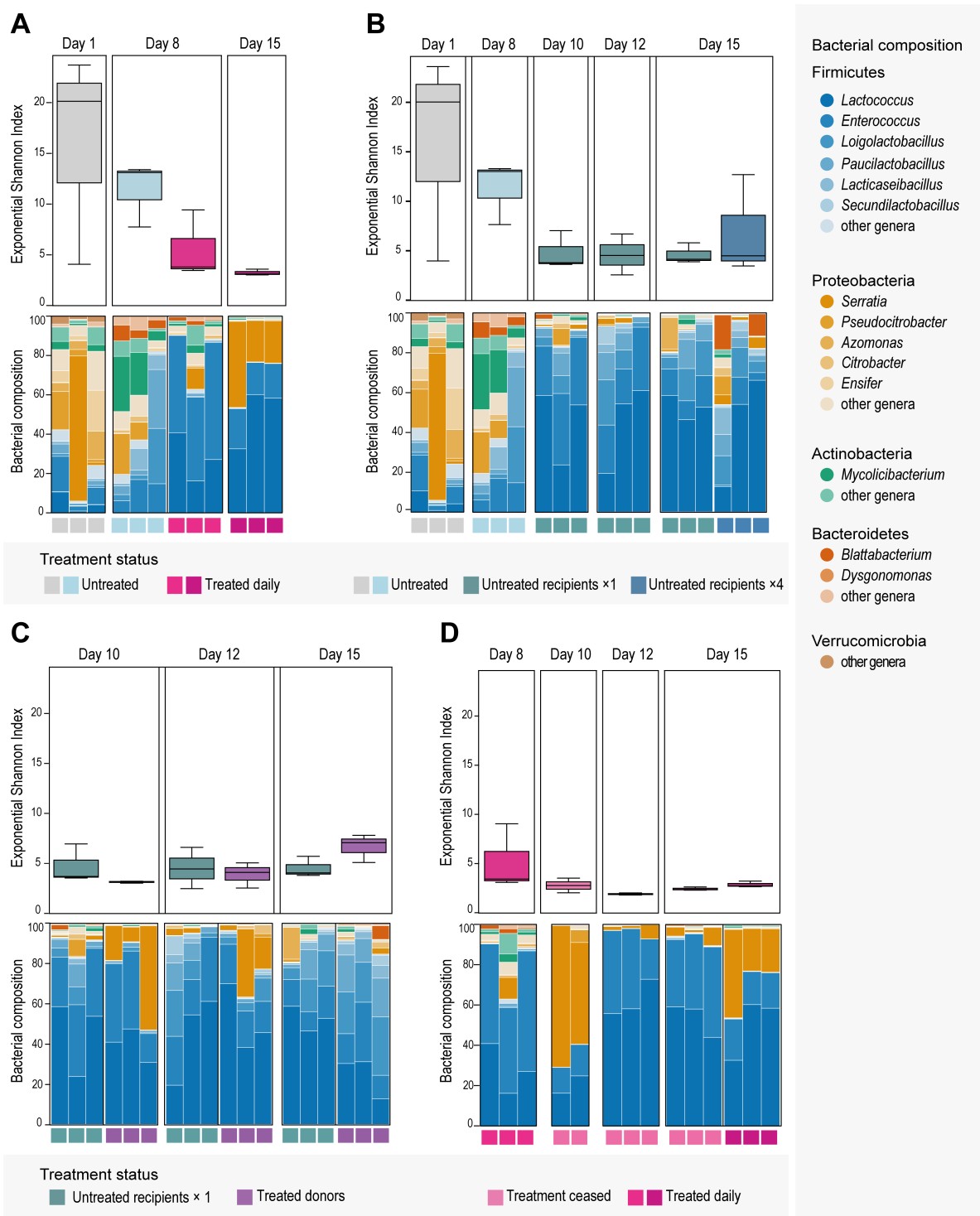

**FIG 4** Bacterial community dynamics in the gut microbiome samples (*n* = 41). Box plots: exponential Shannon diversity index. Box colors indicate treatment status. Bar plots: bacterial Genera composition of the 14 most abundant genera across all gut samples. Genera with low abundance are aggregated in the "other genera" categories, based on their phylum. Bar colors indicate bacterial genera. All triplicates are ordered by day and treatment status. (A) Comparison between untreated and daily-treated microbiomes. (B) Comparison between untreated microbiomes and untreated recipients. The untreated recipients were mixed with treated individuals in one large event, or four smaller events. (C) Comparison between untreated recipients and treated donors, of the one large mixing event. (D) Comparison between daily-treated and post-treatment microbiomes.

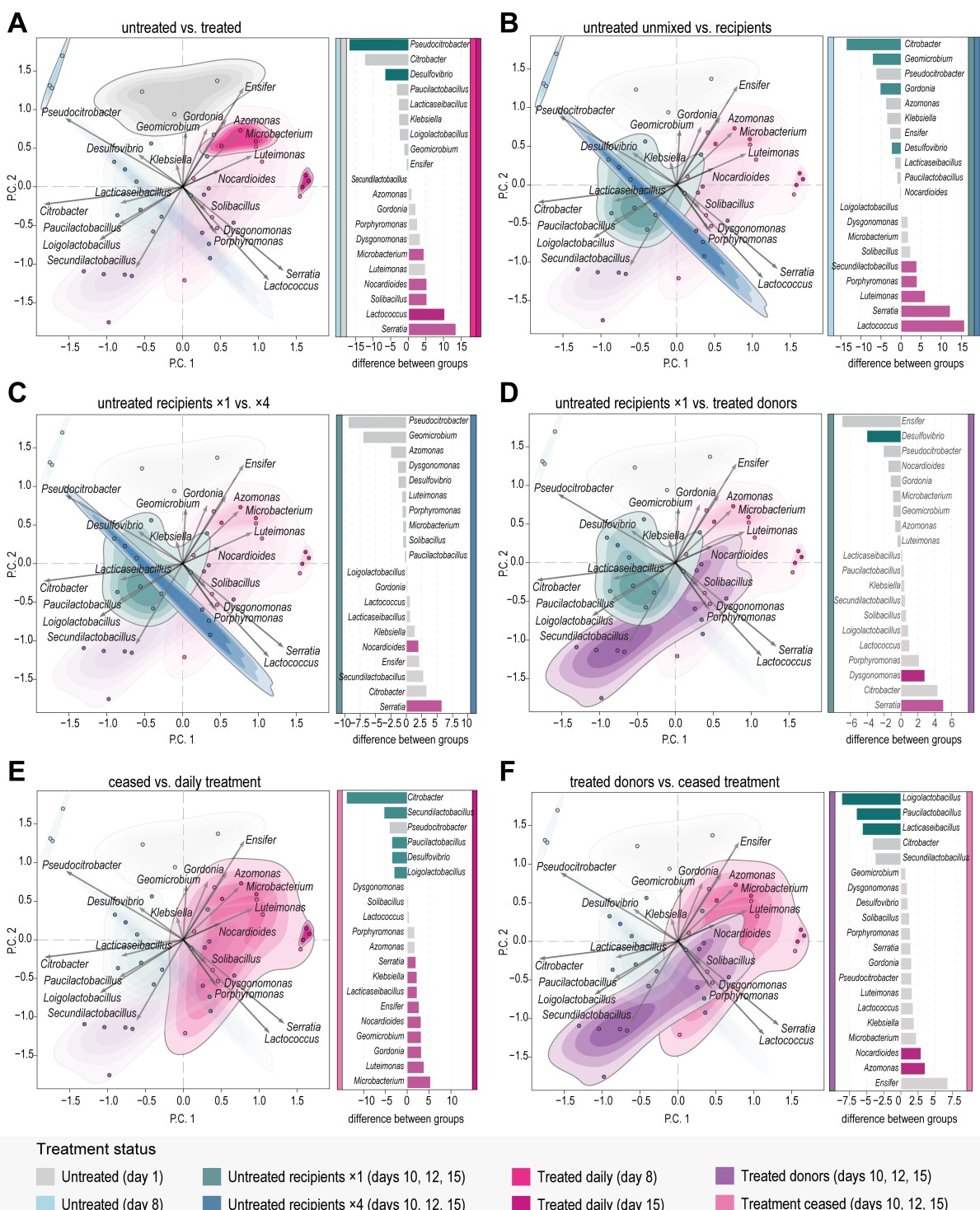

**FIG 5** Differences in bacterial composition between the gut microbiome samples (*n* = 41). Ordination plots: principal component analysis on CLR-transformed bacterial genera fragment counts: P.C. 1 explains 33% of the variation, and P.C. 2 explains 22% of the variation. The same plot is exhibited for (A–F), with different highlights (see Fig. S1B for the non-highlighted version). Point and contour colors indicate the treatment status. Barcharts: Differential abundance of bacterial genera between pairs of groups. Gray bars indicate |effect| < 1, and lighter bars indicate wi.eBH > 0.05. (A) Comparison between untreated (days 1 and 8, *n* = 6) and daily-treated (days 8 and 15, *n* = 6) microbiomes. (B) Comparison between untreated (day 8, *n* = 3) and untreated recipient (day 15, *n* = 6) microbiomes. (C) Comparison between untreated recipients of one large mixing event (day 15, *n* = 3) vs of four small mixing events (day 15, *n* = 3). (D) Comparison between untreated recipients (days 10, 12, and 15, *n* = 9) and treated donors (days 10, 12, and 15, *n* = 9) of the one large mixing event. (E) Comparison between post-treatment (day 15, *n* = 3) and daily-treated microbiomes (day 15, *n* = 3). (F) Comparison between treated donor (days 10, 12, and 15, *n* = 9) and post-treatment microbiomes (days 10, 12, and 15, *n* = 8).

composition. This was confirmed by the differential abundance analysis, where the donors were separated from the daily-treated groups by multiple genera. The donors had a higher abundance of *Citrobacter*, *Pseudocitrobacter*, *Loigolactobacillus*, *Secundilactobacillus*, and *Paucilactobacillus*, whereas the daily-treated microbiomes were more abundant in *Ensifer*, *Microbacterium*, *Gordonia*, *Azomonas*, *Nocardioides*, and *Luteimonas* (Fig. S3B).

The bacterial diversity of the treated microbiomes whose treatment stopped on day 8 remained low until the end of the experiment (Fig. 4D). The samples were diluted in the ordination analysis but remained close to the daily-treated groups and the treated-donor group (Fig. 5E and F). The post-treatment microbiomes were significantly more abundant in *Pseudocitrobacter* and *Citrobacter*, compared to the daily-treated ones, which were more abundant in *Microbacterium*, *Ensifer*, *Gordonia*, and *Azomonas* (Fig. 5E). In contrast, the post-treatment samples had higher abundances of *Azomonas* and *Nocardioides* when compared with the treated donors, who were more abundant in *Loigolactobacillus*, *Paucilactobacillus*, and *Lacticaseibacillus* (Fig. 5F).

## Tracking the ARG dynamics in the soil substrate

### Baseline AMR before treatment

Similar to the gut microbiomes, the ARG level was extremely low in the soil of the untreated terrarium (A) on the 1st and 8th day (Fig. 6A). More than 50% of the ARG content was represented by the gene *aadA*11 that confers resistance to aminoglycosides (Fig. 6A). There were also genes for resistance to tetracycline, *tet*(V) and *tet* (43), folate pathway antagonists (*dfrB*3 and *dfrB*7), macrolide (*ole*(C)), and amphenicol (*cmlV*) (Fig. 6A; Fig. S2B).

### Tetracycline treatment effect was evident in the soil

Tetracycline treatment on the cockroaches had a detectable effect on their soil substrate. There was a notable increase in the tetracycline ARG level in the soil of daily-treated terraria (B, B2) of days 8 and 15 (Fig. 6A). Tetracycline ARGs increased and took a larger part of the ARG composition (Fig. 6A; Fig. S2B). The increase is attributed to the tetracycline ARGs *tet*(S/M) and *tet*(M), which were proved to be differentially abundant (Fig. 6C). These are the same genes that drove the increase in tetracycline ARG in the gut microbiomes (Fig. 3A). The change in ARG composition was also evident in the ordination analysis, where the untreated and treated soil samples separated clearly on the first principal component (Fig. 6C). The flanking region analysis indicated that it is the same *tet*(S/M) and *tet*(M) genes found in the soil as in the treated microbiomes (Fig. S5).

### Transmission of tetracycline AMR was detectable in the soil

The soil substrates that were recipients of treated individuals (terraria A1, B1) exhibited a higher level of tetracycline AMR than those that did not (terrarium A) (Fig. 6A). Tetracycline ARGs increased compositionally (Fig. 6A), with *tet*(S/M) and *tet*(M) being differentially abundant (Fig. 6D). In the ordination analysis, the untreated soil recipients are placed between the untreated and the treated samples (Fig. 6D).

### Minor response of soil bacterial community to tetracycline treatment

The bacterial composition of the soil substrate was distinct from the cockroach gut (Fig. S1C), with a much higher diversity, both in terms of richness and evenness (Fig. S4B). Actinobacteria were the most abundant phylum, followed by Proteobacteria, Firmicutes, and Plantomycetes (Fig. 6B). Several genera were differentially abundant between the soil and the gut microbiomes (Fig. S3C). Yet, the soil microbiome still shared several genera with the gut microbiome, as expected due to the presence of cockroach faeces (e.g., *Enterococcus*, *Lactococcus*, *Paucilactobacillus*, *Azomonas*, and *Microbacterium*) (Fig. 6B).

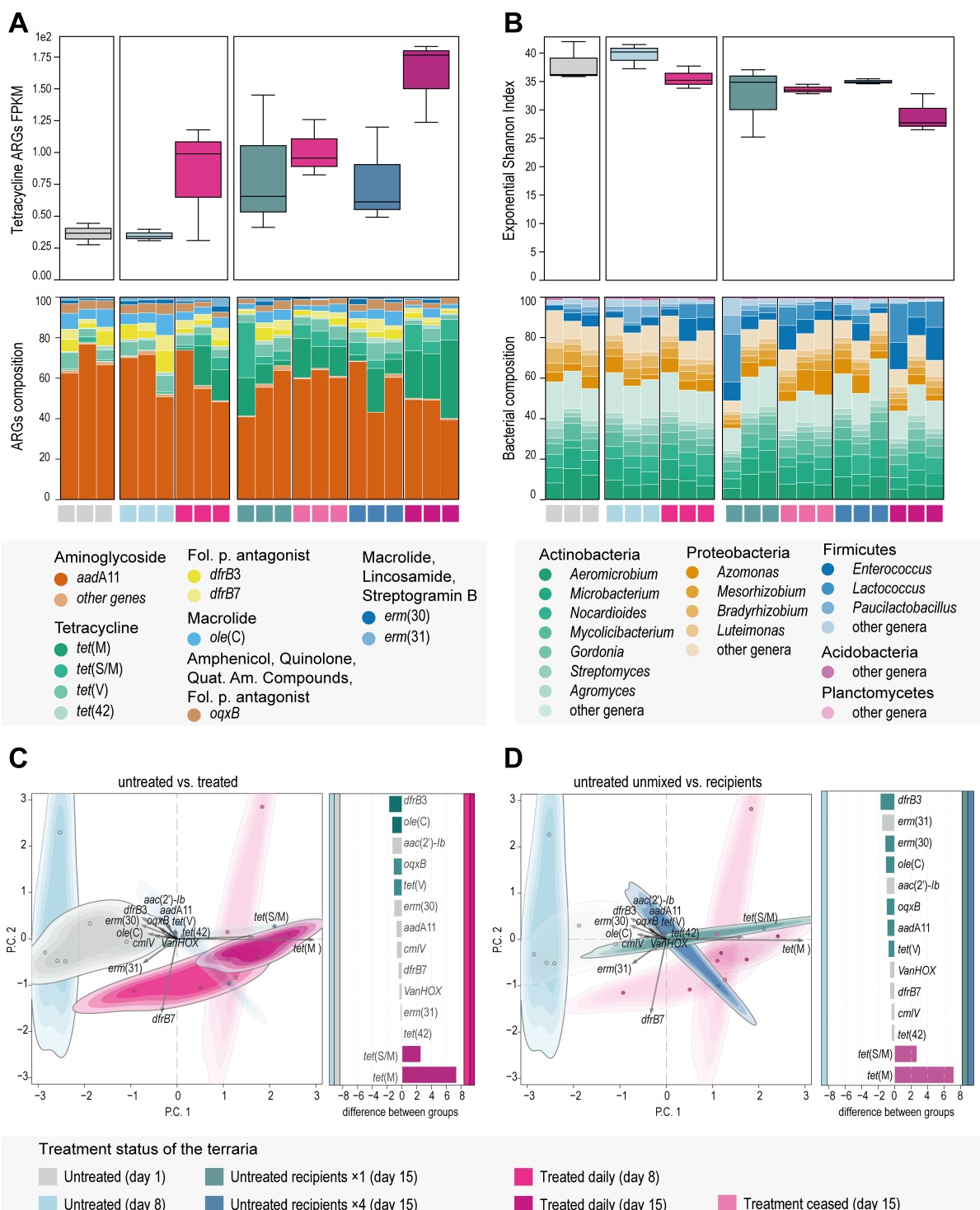

**FIG 6** (A) ARG dynamics in the soil microbiome samples ($n = 21$). Box plots: tetracycline ARGs relative abundance (FPKM). Bar plots: ARG composition of the 11 most abundant ARGs across all soil samples. ARGs with low abundance are aggregated in the "other genes" categories. (B) Bacterial community dynamics in the soil microbiome samples ($n = 21$). Box plots: exponential Shannon diversity index. Bar plots: bacterial genera composition of the 14 most abundant genera across all soil samples. Genera with low abundance are aggregated in the "other genera" categories. (C, D) Differences in ARG composition between the soil microbiome samples. Ordination plots: PCA on CLR-transformed ARGs fragment counts: P.C. 1 explains 60% of the variation, and P.C. 2 explains 14% of the variation. The same plot is exhibited for (C and D), with different highlights (see Fig. S1D for the non-highlighted version). Point and contour colors indicate the treatment status. Barchats: differential abundance of ARGs between pairs of groups. Gray bars indicate |effect| < 1, and lighter bars indicate wi.eBH > 0.05. (C) Comparison between untreated (box A, days 1 and 8, $n = 6$) and daily-treated (box B and B2, days 8 and 15, $n = 6$) soil microbiomes. (D) Comparison between untreated (box A, day 8, $n = 3$) and untreated recipient (box A1 and A2, day 15, $n = 6$) soil microbiomes.

The bacterial diversity of the soil microbiomes decreased after tetracycline treatment (terraria B, B1, B2) and after the introduction of treated donors (A1, A2) (Fig. 6B). In these samples, there was a noticeable increase in Firmicutes, mostly *Enterococcus* and *Lactococcus* (Fig. 6B), which was not statistically significant in the differential abundance analysis. The different groups did not separate adequately in the ordination analysis (Fig. S1D).

## DISCUSSION

While there are plenty of observational studies addressing antimicrobial resistance and its dissemination in complex communities (e.g., see reviews 49–51), there remains a paucity of *in vivo* experimental research on the transmission of ARGs between populations, i.e., on the experimental epidemiology of AMR (8, 15). Here, we challenged live microbiomes in gregarious cockroaches with antibiotics and observed a decrease in bacterial diversity and increase in ARG content in treated microbiomes. We also observed a transmission of those changes between interacting hosts that is spilled over to the previously unchallenged microbiomes.

We conducted our *in vivo* experiments with a gregarious species of cockroach, which are promising experimental animal models for microbiome studies, especially in the field of AMR transmission research (15). Cockroaches harbor a diverse gut microbiome, more diverse than that of the common animal model *D. melanogaster* (52), yet not as complex as cockroaches' close relatives, the eusocial termites (20, 53). Other eusocial insects, in particular the honey bee *Apis mellifera*, carry ARGs (54) and have also been used as experimental models in microbiome research (55). However, the rearing and maintenance of eusocial insect colonies require specialist knowledge (56), while the rigidly structured inter-individual interactions may complicate the dynamics of transmission studies. In contrast, gregarious insects may provide a simpler and more accessible system to test microbial transmission hypotheses. *P. surinamensis* is a gregarious cockroach species, exhibiting a range of interactions between individuals, such as behaviors of moving toward each other, antennal contact and mutual antennations, climbing onto each other (16) and resting in extremely dense groups in the soil substrate (57). Apart from the proximity of the individuals, horizontal (and sometimes vertical) bacterial transmission in gregarious (and social) insects can be attributed to coprophagy and environmental transmission through the shared resources (58–60). Other gregarious cockroach species, such as *B. germanica*, are also a good candidate for microbial transmission studies (15). However, *P. surinamensis* is a parthenogenetic thelytokous species, which means that each colony is clonal (61); this allows for a more accurate detection of disturbance effect on gut microbiomes (e.g., introduction of antibiotics) as it eliminates other host-related factors (23, 62, 63). Overall, gregarious cockroaches are placed well to act as a model for microbiome manipulation studies (e.g., see references 20, 21) with the social structure layer observed in higher animals like humans, unlike most other simple animal models for microbiome research, such as *D. melanogaster,* a predominantly solitary insect (56). Thus, in our study, we demonstrate that *P. surinamensis* is a good candidate for *in vivo* AMR transmission experiments.

Our results on the microbiome composition of the untreated samples are congruent with the findings of the five previous studies on *P. surinamensis* microbiome, which employed amplicon sequencing (18–21, 64). Our study is, to the best of our knowledge, the first one exploring the gut microbiome of *P. surinamensis* with shotgun metagenomic sequencing. Our studies agree that the most abundant phyla were Proteobacteria, Firmicutes, Bacteroidetes, and Actinobacteria, with Synergistetes and Planctomycetes at a lower abundance. Differences in the relative abundance of bacterial genera were found due to the different dietary regimens between the studies. Indeed, diet plays an important role in shaping the gut microbiome composition of *P. surinamensis* (20) and other cockroaches (65, 66). Our study is also the first to examine the ARG content of *P. surinamensis* microbiome. We found that ARGs are present in low abundance in

individuals without any previous exposure to antimicrobials, similar to the results in *B. germanica* (13).

In general, we observed a decrease in the gut microbiome diversity and a change in microbial composition following antimicrobial treatment. This is in agreement with previous *in vivo* studies with *B. germanica*, where antimicrobial treatment also altered the gut bacterial composition (rifampicin [60]; vancomycin and ampicillin [14]; kanamycin [13]). As expected, we observed that antimicrobial treatment resulted in an increased abundance of ARGs, which is in agreement with several previous experimental studies (e.g., see references 13, 67–69).

It is, however, noteworthy that we observed a major increase in resistance among the untreated individuals that were mixed with treated cockroaches. This is in addition to a reduction in bacterial diversity of the untreated recipients, whose microbiome became compositionally more similar to those of the treated donors. This suggests that the gregarious behavior of *P. surinamensis* facilitates the transmission of bacteria between individuals (70, 71) and by extension the transmission of their resistomes. A similar observation has been made among pigs reared together, where the resistomes of treated and untreated individuals converged (5). However, in that study, the cause of convergence was a reduction in ARG abundance in the treated individuals, and not an increase in ARG abundance in the untreated individuals, as in our study. Furthermore, we found that the levels of tetracycline ARGs in both donors and recipients kept increasing until the end of the experiment. This indicates that both antimicrobial treatment and or contact with treated individuals may permanently increase the abundance of ARGs in the gut microbiome (13).

More specifically, we found that the tetracycline resistance genes *tet*(S/M), *tet*(M), *tet*(O), and *tet*(Q) increased after the antimicrobial treatment and were also transmitted to the untreated population after mixing. All four genes encode proteins that protect the bacterial ribosomes from tetracycline (72, 73). The *tet*(M) and tet(S/M) genes have wide host ranges, which is attributed to their association with broad host range conjugative transposons and plasmids (73–76), while tet(O) and *tet*(Q) have been found associated both with the chromosome of some bacterial species and in connection with conjugative transposons and plasmids (72, 73, 77, 78). Unfortunately, without high-quality metagenomic assemblies or long-read sequences, it is not possible to confirm the location of these genes of interest on the chromosomes or the plasmids (5, 79). The wide host range and the association with conjugative transposons could point to horizontal gene transfer during the experiment, which has been recorded before in other cockroaches (e.g.,see reference 80). In the gut samples, the antimicrobial treatment affected both the taxonomic and the ARG composition, yet no notable correlations were found between specific ARGs and bacterial genera. Therefore, further research is required to clarify whether the spread of ARGs is only due to bacterial transmission or also due to horizontal gene transfer.

Interestingly, we also observed that a single large mixing event between untreated and treated cockroaches led to higher levels of resistance compared to smaller and more frequent mixing events. This suggests that one major disturbance has a larger effect than several smaller ones, even though the total cumulative intensity was the same. This is in agreement with previous ecological models showing that both the size and the frequency of a disturbance are important (81) but warrants further studies to show the relative importance and interaction in AMR ecology. This observation is, however, potentially important since it might suggest that if we reduce large transmission events then the normal microbiome might have sufficient resilience to absorb transmission, as recently suggested (82) and in line with our recent modeling approach (9). It was also apparent that when untreated microbiomes were frequently disturbed with small numbers of treated donors, their bacterial diversity was more variable. The smaller magnitude of interactions at each event meant that only a few of the untreated recipients were impacted by the transfer; they were able to replenish their microbiome through interaction with the other untreated individuals during the intervals between

transfers (9, 83, 84). In contrast, this was not possible in the single transfer event, where the majority of untreated recipients came into contact with the donors, resulting in the establishment of similar microbiomes between them.

The soil samples showed the same patterns of increased tetracycline resistance when treating their cockroach populations. Even the soil from the untreated terraria that received treated donors, exhibited increased tetracycline resistance. This similarity between the gut and soil tetracycline ARG levels is likely caused by the presence of cockroach feces in the soil, as previous studies report similarities between the fecal and gut microbiome of cockroaches, which is strengthened by their coprophagy behavior (58–60, 85). Coprophagy facilitates the transmission of bacteria, and potentially of ARGs, between gregarious individuals (59); thus, it is a probable route of ARG transmission between the treated and the untreated individuals in our study. In addition, the similarity between the gut and soil microbiome in our study further confirms that AMR transmission also occurs between microbiomes of different environments (86), especially in light of non-pathogenic bacteria acting as ARG reservoirs with the potential of horizontal transmission to pathogenic bacteria (7, 87). The congruence of the ARG pattern between the soil and gut samples indicates that AMR transmission experiments could also rely exclusively on soil sampling in soil-dwelling animals, avoiding the harvest and dissection of individual cockroaches. This could lead the way toward isolated and undisturbed mesocosm experiments, imitating complex AMR transmission routes in even more controlled conditions.

Experiments on the transmission of AMR are lacking despite the need to validate the findings of several theoretical models (8, 9). It is logistically difficult, time-consuming, and expensive to execute such experiments, especially *in vivo*. Animal models for studying AMR ecology and evolution have mainly been using different mammal species since they reflect the human gut environment most (88); this comes with ethical concerns, financial costs associated with housing and handling the large number of animals needed for statistical analyses, and long time-scales (56). Experiments with gregarious cockroaches provide a solution to this, as they can be easily reared in laboratory conditions, allowing for controlled experimental designs with high reproducibility (15). Their relatively short lifecycle permits timely observations on transmission dynamics. Their small size allows for scaled-up experiments, which can comprise multiple replicates and complex experimental set-ups. Maintaining cockroach colonies requires minimal investment compared to mammalian models, making them a cost-effective choice, particularly for large-scale or long-term studies. Of course, insect models are only the first step in testing modeling hypotheses, as the physiological differences between cockroaches and mammals might result in some variations in microbial interactions and immunity (56). Hence, while cockroaches provide valuable insights into the basic mechanisms of AMR transmission, extrapolating findings to human population dynamics should be approached with caution and complemented by other animal models or clinical observations.

## Conclusions

While cockroaches might not replicate the complexities of human or mammalian microbiomes, they offer a tractable, relevant, and efficient system for studying the dynamics of AMR transmission, especially in the context of densely populated urban environments. Here, we evaluated the effect of antimicrobial treatment and population mixing on the gut microbiome of a gregarious species of cockroaches. Our results showed that *P. surinamensis* live and complex microbiomes respond quickly to stressors, with changes both in ARG content and in the bacterial composition detectable by metagenomics. Tetracycline ARGs were selected for in treated populations and were transmitted to untreated ones upon interaction between them. The effect was greatest when mixing with a larger group of donors only once, compared to smaller groups multiple times, suggesting that both frequency and intensity of microbe transmission affect AMR levels. Further exploration of these findings within the model organism could

include investigating the association between resistance genes and specific bacterial species through metagenomic assemblies and even testing this setup in other model organisms closer related to humans.

## ACKNOWLEDGMENTS

We are grateful to the reviewers for their comments and suggestions to improve the manuscript. We thank Thomas Nordahl Petersen and Baptiste Jacques Philippe Avot for assisting with the bioinformatic analyses and Patrick Munk for his assistance with the AMR gene homology reduction. We are grateful to Philip T. L. C. Clausen and Nikiforos G. Pyrounakis for the flanking region analysis.

The work is supported by The Novo Nordisk Foundation (NNF16OC0021856: Global Surveillance of Antimicrobial Resistance) and the European Union's Horizon 2020 research and innovation program under VEO grant agreement No. 874735.

A.B., E.E.B.J., S.O., F.M.A., and C.B. contributed to conception and design of the study. A.B., E.E.B.J., and A.V.B. performed the experiments. E.E.B.J. ran the bioinformatic analyses. A.B. and E.E.B.J. carried out the analyses, C.B. contributed to the analyses. A.B. created the visualizations and wrote the manuscript. E.E.B.J., A.V.B., C.B., S.O., and F.M.A. commented on the manuscript. All authors contributed to manuscript revision, read, and approved the submitted version.

## AUTHOR AFFILIATION

[1]Research Group for Genomic Epidemiology, Technical University of Denmark, Kgs., Lyngby, Denmark

## AUTHOR ORCIDs

Amalia Bogri http://orcid.org/0000-0002-9127-6146
Emilie Egholm Bruun Jensen http://orcid.org/0000-0002-3214-7918
Asbjørn Vedel Borchert http://orcid.org/0009-0007-0961-645X
Christian Brinch http://orcid.org/0000-0002-5074-7183
Saria Otani http://orcid.org/0000-0002-2538-8086
Frank M. Aarestrup http://orcid.org/0000-0002-7116-2723

## FUNDING

| Funder | Grant(s) | Author(s) |
|---|---|---|
| Novo Nordisk Fonden (NNF) | NNF16OC0021856 | Amalia Bogri |
| Horizon 2020 (EU): VEO | No. 874735 | Amalia Bogri |

## AUTHOR CONTRIBUTIONS

Amalia Bogri, Conceptualization, Data curation, Formal analysis, Investigation, Methodology, Software, Visualization, Writing – original draft, Writing – review and editing | Emilie Egholm Bruun Jensen, Conceptualization, Data curation, Formal analysis, Investigation, Methodology, Software, Writing – review and editing | Asbjørn Vedel Borchert, Investigation, Methodology, Writing – review and editing | Christian Brinch, Conceptualization, Methodology, Software, Supervision, Writing – review and editing | Saria Otani, Conceptualization, Data curation, Methodology, Project administration, Supervision, Writing – review and editing | Frank M. Aarestrup, Conceptualization, Funding acquisition, Investigation, Methodology, Resources, Supervision, Writing – review and editing

## DATA AVAILABILITY

All metagenomics sequences from all gut and soil microbiomes are submitted to the European Nucleotide Archive (ENA) with the following project accession number: PRJEB66261.

## ADDITIONAL FILES

The following material is available online.

### Supplemental Material

**Supplemental Figures (mSystems01018-23-s0001.pdf).** Figures S1-S5.

### Open Peer Review

**PEER REVIEW HISTORY (review-history.pdf).** An accounting of the reviewer comments and feedback.

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
