## [Reviewer comments · mSystems]

Transmission of antimicrobial resistance in the gut microbiome of gregarious cockroaches: the importance of interaction between antibiotic exposed and non-exposed populations

Amalia Bogri, Emilie Jensen, Asbjørn Borchert, Christian Brinch, Saria Otani, and Frank Aarestrup

Corresponding Author(s): Amalia Bogri, Danmarks Tekniske Universitet

Review Timeline:

Submission Date:	September 21, 2023
Editorial Decision:	October 30, 2023
Revision Received:	November 14, 2023
Accepted:	November 17, 2023

Editor: Jonathan Klassen

Reviewer(s): Disclosure of reviewer identity is with reference to reviewer comments included in decision letter(s). The following individuals involved in review of your submission have agreed to reveal their identity: Aram Mikaelyan (Reviewer #1)

Transaction Report:

DOI: <https://doi.org/10.1128/msystems.01018-23>

Re: mSystems01018-23 (Transmission of antimicrobial resistance in the gut microbiome of sub-social cockroaches: the importance of interaction between antibiotic exposed and non-exposed populations)

Dear Ms. Amalia Bogri:

In your revision, please particularly note the concerns raised by the reviewers regarding the justification of the model system with regard to the study of antimicrobial resistance.

Revision Guidelines

Sincerely,
Jonathan Klassen
Editor
mSystems

Reviewer #1 (Comments for the Author):

The authors of this manuscript sought to address the knowledge gap surrounding the transmission dynamics of antimicrobial resistance (AMR), particularly via the gut microbiome. Recognizing the limited in vivo experimental studies on this topic, they utilized *Pycnoscelus surinamensis* as a model to study the resistome. Through their investigation, they attempted to highlight the

significant role of the microbiome in AMR transmission and the potential implications for broader ecosystems. Their main conclusions emphasize the importance of understanding horizontal gene transfer mechanisms and the need for further in-depth research on insect models.

While this research holds potential and provides intriguing insights into AMR transmission via the gut microbiome of *Pycnoscelus surinamensis*, several concerns emerge. Overall, I think the exact hypotheses that ARG researchers could test with *P. surinamensis* that they cannot with other model systems should be made clear. I expand on these concerns below:

Model Organism Selection: The authors need to provide more context to their choice of *Pycnoscelus surinamensis* as a model organism. *P. surinamensis* is a litter-feeding, parthenogenic cockroach species that has a minor pest status in some countries. It has a complex microbiome, but so does a species like the German cockroach, *Blattella germanica*, notorious for transmitting human pathogens and antibiotic resistance, might have been a more relevant pick for simulating "human-associated" microbiomes. Cockroaches, by nature, partake in coprophagy, enabling them to effectively transmit microbiome components and resistance, arguably more so than many mammals and other insects. This aspect, which the authors use to justify soil microbiome sampling, merits a broader discussion, especially concerning ARG transmission dynamics between treated and untreated groups. Furthermore, *Pycnoscelus*' parthenogenetic nature complicates its selection as a straightforward model, a point not addressed in relation to its potential implications on the study's outcomes or its use as a model system for other questions.

AMR Dynamics: The authors' assertion that there are "limited existing studies on AMR transmission dynamics" may be misleading. Insects like house flies have been extensively researched for their role in disseminating antibiotic resistance and pathogens. Given their shorter generational periods compared to *Pycnoscelus*, flies could be better models for such investigations. Again, it's important that the authors expound on why they favored *Pycnoscelus* over other potential candidates and offer a clear understanding of the models they sought to test or replicate.

Terminological Inconsistencies: The manuscript's interchangeable use of "semi-social," "sub-social," and "gregarious" to characterize *Pycnoscelus surinamensis* is concerning. It's imperative to note that these aren't mere descriptive terms but hold specific significance in entomology and animal behavior. While *Pycnoscelus* is gregarious, it doesn't fit the established mold for subsocial or semisocial behaviors. Semisociality, as seen in Halictid "sweat" bees, typically denotes species where members of the same generation partake in brood care but have distinct reproductive roles. Contrastingly, subsociality, observed in cockroach species like *Salganea*, and *Cryptocercus*, pertains to a parent (typically the mother) caring for her brood without a clear division in reproductive responsibilities. Misusing these terms risks misleading readers about *Pycnoscelus*'s natural behaviors and the study's implications.

In closing, the study is undeniably informative, but addressing the aforementioned concerns will amplify its clarity, context, and relevance, if it is to stand as a robust contribution to the scientific discussion on AMR transmission dynamics.

Minor comments:

Pg 2: What do you mean by "microbiome-rich"?

Pg 2: "we show that resistance increased in untreated populations after interacting with the treated ones." - I am trying to understand how this study pushes forward our understanding of AMR-resistance. Hasn't this particular result (not in roaches) already been shown in pigs by Tams et al. 2023.

Pg 4: Cockroaches don't "shed their gut" - just their gut lining or the "gut cuticle"

Pg 4: Who was the breeder the cockroaches were procured from?

Pg 7: It is clear that the authors have thought about mortality in the context of Tetracycline doses, but the antibiotic has been shown to directly impact mitochondrial function in *Drosophila*. Can the authors comment on its potential effects in cockroaches?

Pg 23: I would prefer it if the authors clarified where exactly the "paucity of in vivo experimental research" is, in order to streamline the focus of the paper. One could easily argue that there is considerable in vivo research that has been done in this field from the labs of Soren Sorenson or Julian Marchesi, to name a few.

Pg 23: "While not eusocial, *P. surinamensis* is a gregarious, sub-social cockroach species, exhibiting a high level of interactions between individuals." - what do you mean by a "high level of interactions"? Are there any current studies supporting the subsocial status of *Pycnoscelus*?

Pg 23: Correct spelling: "gregarius"

Reviewer #2 (Comments for the Author):

The authors have performed a captivating and well-designed study to shed light on AMR transmission and the gut microbiome and environment roles using a semi-social species of cockroaches as a model. These types of studies are scarce, and they are relevant. Overall, the article is clear and brings new knowledge on using insects as models for answering ecological questions, including AMR dynamics. I only have some minor comments to improve the manuscript.

Tetracycline resistance genes are often found on transmissible elements as plasmids. Where are the investigated tetracycline ARGs localized? Plasmids and/or chromosomes? Did the authors perform some analyses on plasmid reads? Although bacteria can be transmitted among microbiome, plasmids could also play a role in the increase in ARGs. Also, besides the selection of resistant bacteria by the antibiotics, exchange in plasmids could also contribute to explaining the rise in antimicrobial resistance.

"The number of aligned fragments of each gene was adjusted for their ResFinder reference template length, by dividing by the length of the reference gene and multiplying with 10³". It needs to be clarified why the authors multiply by 1000. Please explain it.

"We also binned the ARG variants to the level of drug class, as in Munk et al., 2022." Is it perhaps drug class?

"The generic taxonomic level." Is that the genus level?

We are grateful for the suggestions of the two reviewers that greatly improved our manuscript. Below, we respond point-by-point to the reviewers' comments.

The reviewers' comments are in **blue**, whereas our responses in **black**.

Reviewer #1 (Comments for the Author):

The authors of this manuscript sought to address the knowledge gap surrounding the transmission dynamics of antimicrobial resistance (AMR), particularly via the gut microbiome. Recognizing the limited in vivo experimental studies on this topic, they utilized *Pycnoscelus surinamensis* as a model to study the resistome. Through their investigation, they attempted to highlight the significant role of the microbiome in AMR transmission and the potential implications for broader ecosystems. Their main conclusions emphasize the importance of understanding horizontal gene transfer mechanisms and the need for further in-depth research on insect models.

While this research holds potential and provides intriguing insights into AMR transmission via the gut microbiome of *Pycnoscelus surinamensis*, several concerns emerge. Overall, I think the exact hypotheses that ARG researchers could test with *P. surinamensis* that they cannot with other model systems should be made clear. I expand on these concerns below:

Model Organism Selection:

The authors need to provide more context to their choice of *Pycnoscelus surinamensis* as a model organism. *P. surinamensis* is a litter-feeding, parthenogenic cockroach species that has a minor pest status in some countries. It has a complex microbiome, but so does a species like the German cockroach, *Blattella germanica*, notorious for transmitting human pathogens and antibiotic resistance, might have been a more relevant pick for simulating "human-associated" microbiomes. Cockroaches, by nature, partake in coprophagy, enabling them to effectively transmit microbiome components and resistance, arguably more so than many mammals and other insects. This aspect, which the authors use to justify soil microbiome sampling, merits a broader discussion, especially concerning ARG transmission dynamics between treated and untreated groups. Furthermore, *Pycnoscelus*' parthenogenetic nature complicates its selection as a straightforward model, a point not addressed in relation to its potential implications on the study's outcomes or its use as a model system for other questions.

We agree with the reviewer that multiple organisms might have served the same purpose to carry out an *in vivo* experiment, not solely our chosen model, *Pycnoscelus surinamensis*. Therefore, we have rephrased several sentences throughout the manuscript to be more inclusive to other gregarious cockroach species, and particularly *B. germanica*.

We expand on our choice of a gregarious cockroach species in the second paragraph of the Discussion. In there, we also discuss the parthenogenetic nature of *P. surinamensis* and the possible benefits from using a clonal colony. Thus, we rewrote the 2nd paragraph of the Discussion completely, adding several new references, as follows (Marked-up manuscript, pages 15-16):

‘We conducted our *in vivo* experiments with a gregarious species of cockroach, which are promising experimental animal models for microbiome studies, especially in the field of AMR transmission research (Llop et al., 2018). Cockroaches, harbour a diverse gut microbiome, more diverse than that of the common animal model *Drosophila melanogaster* (Ourry et al., 2020), yet not as complex as cockroaches’ close relatives, the eusocial termites (Inward et al., 2007; Richards et al., 2017). Other eusocial insects, in particular the honey bee *Apis mellifera*, carry ARGs (Sun et al., 2023) and have also been used as experimental models in microbiome research (Zheng et al., 2018). However, the rearing and maintenance of eusocial insect colonies require specialist knowledge (Douglas, 2019), while the rigidly structured inter-individual interactions may complicate the dynamics of transmission studies. In contrast, gregarious insects may provide a simpler and more accessible system to test microbial transmission hypotheses. *P. surinamensis* is a gregarious, cockroach species, exhibiting a range of interactions between individuals, such as behaviours of moving towards each other, antennal contact and mutual antennations, climbing onto each other (Legendre et al., 2014) and resting in extremely dense groups in the soil substrate (Laurent-Salazar et al., 2019). Apart from the proximity of the individuals, horizontal (and sometimes vertical) bacterial transmission in gregarious (and social) insects can be attributed to coprophagy and environmental transmission through the shared resources (Jahnes et al., 2018; Onchuru et al. 2018; Rosas et al., 2018). Other gregarious cockroach species, such as *Blattella germanica*, are also a good candidate for microbial transmission studies (Llop et al., 2018). However, *P. surinamensis* is a parthenogenetic thelytokous species, which means that each colony is clonal (Zangl et al., 2019); this allows for a more accurate detection of disturbance effect on gut microbiomes (e.g. introduction of antibiotics) as it eliminates other host-related factors (Ebert, 2022; Vujkovic-Cvijin et al., 2020; Kurilshikov et al., 2021). Overall, gregarious

cockroaches are placed well to act as a model for microbiome manipulation studies (e.g., Mikaelyan et al., 2015; Richards et al., 2017) with the social structure layer observed in higher animals like humans, unlike most other simple animal models for microbiome research, such as *D. melanogaster*, a predominantly solitary insect (Douglas, 2019). Thus, in our study, we demonstrate that *P. surinamensis* is a good candidate for *in vivo* AMR transmission experiments.’

We also note we chose to work with *P. surinamensis* for practical reasons; some of the authors already had experience rearing *P. surinamensis* populations, and conducting experiments with this species, as well as *P. surinamensis* is easily procured from commercial breeders in the EU (in this case, blattaria.fr).

Regarding the coprophagous behaviour and the soil sampling, we also expanded our Discussion (Marked-up manuscript, page 19) as follows: ‘This similarity between the gut and soil tetracycline ARG levels is likely caused by the presence of cockroach faeces in the soil, as previous studies report similarities between the faecal and gut microbiome of cockroaches, which is strengthened by their coprophagy behaviour (Jahnes et al., 2018; Onchuru et al., 2018; Kakumanu et al., 2018; Rosas et al., 2018). Coprophagy facilitates the transmission of bacteria, and potentially of ARGs, between gregarious individuals (Onchuru et al., 2018); thus, it is a probable route of ARG transmission between the treated and the untreated individuals in our study. In addition, the similarity between the gut and soil microbiome in our study further confirms that AMR transmission also occurs between microbiomes of different environments (Sessitsch et al., 2023), especially in light of non-pathogenic bacteria acting as ARG reservoirs with the potential of horizontal transmission to pathogenic bacteria (Holmes, 2016; Muñoz-Benavent et al., 2021).’

AMR Dynamics:

The authors' assertion that there are "limited existing studies on AMR transmission dynamics" may be misleading. Insects like house flies have been extensively researched for their role in disseminating antibiotic resistance and pathogens. Given their shorter generational periods compared to *Pycnoscelus*, flies could be better models for such investigations. Again, it's important that the authors expound on why they favored *Pycnoscelus* over other potential candidates and offer a clear understanding of the models they sought to test or replicate.

We understand the concern of the reviewer regarding our statement that there are "limited existing studies on AMR transmission dynamics" in our Introduction. Indeed, there are several studies examining the dissemination of AMR between the environment, animals, and humans (e.g. reviews by Huijbers et al., 2015; Ikhimiukor et al., 2022; Rawat et al., 2023). Usually, such studies are *observational*, and often concern AMR transmission between different animal species. Especially Diptera (e.g. flies and mosquitos) have been extensively sampled for their dissemination of AMR and of pathogens to humans and farm animals (e.g. Onwugamba et al., 2018; Fukuda et al., 2019).

Our intention was to highlight the scarcity of *experimental* studies on AMR transmission dynamics. We have corrected our phrasing regarding this across the entire manuscript, and particularly in the first and second paragraph of the Discussion (Marked-up manuscript, pages 15-16). On that note, (house) flies are usually studied as vectors of AMR to other animals/humans/environments, and they are not common experimental models for AMR transmission between individuals of their own population/species. This could be because they do not exhibit the high levels of gregariousness reported in cockroaches, and are described as predominantly solitary.

Terminological Inconsistencies:

The manuscript's interchangeable use of "semi-social," "sub-social," and "gregarious" to characterize *Pycnoscelus surinamensis* is concerning. It's imperative to note that these aren't mere descriptive terms but hold specific significance in entomology and animal behavior. While *Pycnoscelus* is gregarious, it doesn't fit the established mold for subsocial or semisocial behaviors. Semisociality, as seen in Halictid "sweat" bees, typically denotes species where members of the same generation partake in brood care but have distinct reproductive roles. Contrastingly, subsociality, observed in cockroach species like *Salganea*, and *Cryptocercus*, pertains to a parent (typically the mother) caring for her brood without a clear division in reproductive responsibilities. Misusing these terms risks misleading readers about *Pycnoscelus*'s natural behaviors and the study's implications.

We agree with the reviewer that we should standardise the terminology. Indeed, there have been no reports of parental care (sub-social) or alloparental care and reproductive division of labor (semi-social) in *P. surinamensis*. Therefore, we have replaced all mentions of 'sub-social' and 'semi-social' with the appropriate term, 'gregarious', across the entire manuscript.

In closing, the study is undeniably informative, but addressing the aforementioned concerns will amplify its clarity, context, and relevance, if it is to stand as a robust contribution to the scientific discussion on AMR transmission dynamics.

Minor comments:

Pg 2: What do you mean by "microbiome-rich"?

We meant that *P. surinamensis*, as a cockroach, has a complex/diverse microbiome. We agree that the term 'microbiome-rich' is not correct, and we removed it (Marked-up manuscript, page 2).

Pg 2: "we show that resistance increased in untreated populations after interacting with the treated ones." - I am trying to understand how this study pushes forward our understanding of AMR-resistance. Hasn't this particular result (not in roaches) already been shown in pigs by Tams et al. 2023.

We agree with the reviewer that this is unclear; we did not mention the difference between our findings and Tams' in our discussion. This particular result was *not* shown in Tams et al. (2023), so we did not change our sentence in page 2.

In Tams et al. (2023) the resistance gene abundance indeed converged between the treated and untreated pigs by week 14. This happened because the ARG abundance of the treated pigs decreased (see Tams Results page 8 and Figure 6), and not because AMR levels increased in the untreated pigs. They also discuss the decrease of ARGs in the treated pigs during the experiment (see Tams Discussion, page 13), but not the increase of ARGs in the untreated pigs. We showed that AMR levels increased in the untreated cockroaches after contact with the treated cockroaches (our Figure 2B). We have added the comparison with Tams et al. (2023) to our discussion (Marked-up manuscript, page 17), as follows:

'A similar observation has been made among pigs reared together, where the resistomes of treated and untreated individuals converged (Tams et al., 2023). However, in Tams et al. (2023), the cause of convergence was a reduction in ARG abundance in the treated individuals, and not an increase in ARG abundance in the untreated individuals, as in our study.'

Pg 4: Cockroaches don't "shed their gut" - just their gut lining or the "gut cuticle"

We changed the phrases ‘shed their gut’ and ‘shed their digestive tract completely’ to ‘shed their gut lining’ (Marked-up manuscript, page 4).

Pg 4: Who was the breeder the cockroaches were procured from?

We included the name of the commercial breeder: ‘blattaria.fr’ (Marked-up manuscript, page 4).

Pg 7: It is clear that the authors have thought about mortality in the context of Tetracycline doses, but the antibiotic has been shown to directly impact mitochondrial function in *Drosophila*. Can the authors comment on its potential effects in cockroaches?

To address the potential effects of tetracycline in cockroaches, we added the following text (Marked-up manuscript, pages 6-7):

‘Tetracyclines have been found to inhibit not only bacteria but also the mitochondria in eukaryotic cells; specifically, in *Drosophila melanogaster* and *Blattella germanica*, tetracyclines delayed and reduced growth and fecundity, and increased mobility early in life (Moullan et al., 2015; Llop et al., 2018). The duration of our pilot and our main experiments was short (two weeks). Thus, as we sampled late stage juveniles that did not moult during the experiment, we are confident that delays in growth and fecundity would not affect our results. Tetracycline has the potential to affect the mobility of flies by improving muscle fitness, but this effect was only minor later in life (Moullan et al., 2015). We did not perform mobility assays on *P. surinamensis*, and, to our knowledge, tetracycline’s effect on mobility has not been explored in hemimetabolous insects. Thus, we sampled only late stage juveniles, expecting that their mobility is less or not affected by tetracycline.’

Pg 23: I would prefer it if the authors clarified where exactly the "paucity of in vivo experimental research" is, in order to streamline the focus of the paper. One could easily argue that there is considerable in vivo research that has been done in this field from the labs of Soren Sorenson or Julian Marchesi, to name a few.

We wanted to highlight that there is a scarcity in experimental *in vivo* research in AMR transmission, as opposed to observational studies. We rephrased the sentence (Marked-up manuscript, page 15) as:

‘While there are plenty of observational studies addressing antimicrobial resistance and its dissemination in complex communities (e.g. see reviews Huijbers et al., 2015; Ikhimiukor et al., 2022; Rawat et al., 2023), there remains a paucity of *in vivo* experimental

research on the transmission of ARGs between populations, i.e. on the experimental epidemiology of AMR (Llop et al., 2018; Blanquart, 2019).’

We could not find *in vivo* (in animals) experimental research on gut microbiome and AMR in Søren J. Sørensen’s (or Soren Sorenson) research. We included a relevant observational study on AMR of his. Likewise, we could also not identify any *in vivo* (in animals) experimental research on gut microbiome and AMR in Julian Marchesi’s research, as he is mostly working with cases of clinical relevance in humans.

Pg 23: "While not eusocial, *P. surinamensis* is a gregarius, sub-social cockroach species, exhibiting a high level of interactions between individuals." - what do you mean by a "high level of interactions"? Are there any current studies supporting the subsocial status of *Pycnoscelus*?

We agree that there is no evidence that there is any parental care; thus, *Pycnoscelus* is not sub-social. We have changed this across the entire manuscript and title from ‘sub-social’ (and ‘semi-social’) to ‘gregarious’.

We also changed the phrase ‘exhibiting a high level of interactions between individuals’ with ‘exhibiting a range of interactions between individuals, such as behaviours of moving towards each other, antennal contact and mutual antennations, climbing onto each other (Legendre et al., 2014) and resting in extremely dense groups in the soil substrate (Laurent-Salazar et al., 2019)’ (Marked-up manuscript, page 16).

Pg 23: Correct spelling: "gregarius"

We corrected the spelling to ‘gregarious’.

Reviewer #2 (Comments for the Author):

The authors have performed a captivating and well-designed study to shed light on AMR transmission and the gut microbiome and environment roles using a semi-social species of cockroaches as a model. These types of studies are scarce, and they are relevant. Overall, the article is clear and brings new knowledge on using insects as models for answering ecological questions, including AMR dynamics. I only have some minor comments to improve the manuscript.

Tetracycline resistance genes are often found on transmissible elements as plasmids. Where are the investigated tetracycline ARGs localized? Plasmids and/or chromosomes? Did the authors perform some analyses on plasmid reads? Although bacteria can be transmitted among microbiome, plasmids could also play a role in the increase in ARGs. Also, besides the selection of resistant bacteria by the antibiotics, exchange in plasmids could also contribute to explaining the rise in antimicrobial resistance.

We agree with the reviewer, and we provide more explanation on the transmission of ARGs in our Discussion (Marked-up manuscript, page 18) as follows:

‘More specifically, we found that the tetracycline resistance genes *tet(S/M)*, *tet(M)*, *tet(O)* and *tet(Q)* increased after the antimicrobial treatment and were also transmitted to the untreated population after mixing. All four genes encode proteins that protect the bacterial ribosomes from tetracycline (Leng et al., 1997; Roberts, 2005). The *tet(M)* and *tet(S/M)* genes have wide host ranges, which is attributed to their association with broad host range conjugative transposons and plasmids (Ciric et al., 2014; Roberts, 2005, Barile et al., 2012; Lo et al., 2019), while *tet(O)* and *tet(Q)* have been found associated both with the chromosome of some bacterial species and in connection with conjugative transposons and plasmids (Leng et al., 1997, Roberts, 2005; Dasti et al., 2007; Crespo et al., 2012). Unfortunately, without high quality metagenomic assemblies or long read sequences it is not possible to confirm the location of these genes of interest on the chromosomes or the plasmids (Johansson et al., 2023; Tams et al., 2023). The wide host range and the association with conjugative transposons could point to horizontal gene transfer during the experiment, which has been recorded before in other cockroaches (e.g. Anacarso et al., 2016). In the gut samples, the antimicrobial treatment affected both the taxonomic and the ARG composition, yet no notable correlations were found between specific ARGs and bacterial genera. Therefore, further research is required to clarify whether the spread of ARGs is only due to bacterial transmission or also due to horizontal gene transfer.’

"The number of aligned fragments of each gene was adjusted for their ResFinder reference template length, by dividing by the length of the reference gene and multiplying with 10³". It needs to be clarified why the authors multiply by 1000. Please explain it.

We adjusted by the length of the reference gene in kilobases, hence the multiplying by 1000. This is a common reference length adjustment choice (e.g. Munk et al., 2022). We corrected the sentence as: ‘The number of aligned fragments of each gene was adjusted for their

ResFinder reference template length, by dividing by the length of the reference gene (in kilobases)' (Marked-up manuscript, page 9).

"We also binned the ARG variants to the level of drug class, as in Munk et al., 2022." Is it perhaps drug class?

It is indeed drug class, and the spelling mistake is now corrected (Marked-up manuscript, page 9).

"The generic taxonomic level." Is that the genus level?

Yes, it is the genus level, so we corrected the phrase to 'at the taxonomic level of genus' (Marked-up manuscript, page 9).

In implementing the changes suggested by both reviewers, we have included the following new references in the manuscript:

1. Aires J, Doucet-Populaire F, Butel MJ. 2007. Tetracycline Resistance Mediated by tet (W), tet (M), and tet (O) Genes of Bifidobacterium Isolates from Humans. *Appl Environ Microbiol* 73:2751–2754.
2. Anacarso I, Iseppi R, Sabia C, Messi P, Cond C, Bondi M, Niederhusern S de. 2016. Conjugation-Mediated Transfer of Antibiotic-Resistance Plasmids Between Enterobacteriaceae in the Digestive Tract of *Blaberus craniifer* (Blattodea: Blaberidae). *J Méd Èntomol* 53:591–597.
3. Barile S, Devirgiliis C, Perozzi G. 2012. Molecular characterization of a novel mosaic tet(S/M) gene encoding tetracycline resistance in foodborne strains of *Streptococcus bovis*. *Microbiology* 158:2353–2362.
4. Burow E, Käsbohrer A. 2017. Risk Factors for Antimicrobial Resistance in *Escherichia coli* in Pigs Receiving Oral Antimicrobial Treatment: A Systematic Review. *Microb Drug Resist* 23:194–205.
5. Ciric L, Brouwer MSM, Mullany P, Roberts AP. 2014. Minocycline resistance in an oral *Streptococcus infantis* isolate is encoded by tet(S) on a novel small, low copy number plasmid. *FEMS Microbiol Lett* 353:106–115.
6. Crespo MD, Olson JW, Altermann E, Siletzky RM, Kathariou S. 2012. Chromosomal tet (O)-Harboring Regions in *Campylobacter coli* Isolates from Turkeys and Swine. *Appl Environ Microbiol* 78:8488–8491.

7. Dasti JI, Groß U, Pohl S, Lugert R, Weig M, Schmidt-Ott R. 2007. Role of the plasmid-encoded tet(O) gene in tetracycline-resistant clinical isolates of *Campylobacter jejuni* and *Campylobacter coli*. *J Méd Microbiol* 56:833–837.
8. Ebert D. 2022. *Daphnia* as a versatile model system in ecology and evolution. *EvoDevo* 13:16.
9. Huijbers PMC, Blaak H, Jong MCM de, Graat EAM, Vandenbroucke-Grauls CMJE, Husman AM de R. 2015. Role of the Environment in the Transmission of Antimicrobial Resistance to Humans: A Review. *Environ Sci Technol* 49:11993–12004.
10. Ikhimiukor OO, Odih EE, Donado-Godoy P, Okeke IN. 2022. A bottom-up view of antimicrobial resistance transmission in developing countries. *Nat Microbiol* 7:757–765.
11. Jahnes BC, Herrmann M, Sabree ZL. 2019. Conspecific coprophagy stimulates normal development in a germ-free model invertebrate. *Peerj* 7:e6914.
12. Johansson MHK, Aarestrup FM, Petersen TN. 2023. Importance of mobile genetic elements for dissemination of antimicrobial resistance in metagenomic sewage samples across the world. *PLOS ONE* 18:e0293169.
13. Kurilshikov A, Medina-Gomez C, Bacigalupe R, Radjabzadeh D, Wang J, Demirkan A, Roy CIL, et al. 2021. Large-scale association analyses identify host factors influencing human gut microbiome composition. *Nat Genet* 53:156–165.
14. Laurent-Salazar, MO., Bouchebti, S., Lihoreau, M. (2019). Gregarious Cockroaches. In: Starr, C. (eds) *Encyclopedia of Social Insects*. Springer, Cham.
15. Leng Z, Riley DE, Berger RE, Krieger JN, Roberts MC. 1997. Distribution and mobility of the tetracycline resistance determinant tetQ. *J Antimicrob Chemother* 40:551–559.
16. Li X, Rensing C, Vestergaard G, Arumugam M, Nesme J, Gupta S, Brejnrod AD, Sørensen SJ. 2022. Metagenomic evidence for co-occurrence of antibiotic, biocide and metal resistance genes in pigs. *Environ Int* 158:106899.
17. Lima B. M. F. V., Moreira J. O. T., dos Santos J. C., dos Santos J. B. C. (2012) Biology and control of *Pycnoscelus surinamensis* L. for vegetable extracts and commercial entomopathogenic fungi, *Revista Caatinga*, Mossoró 25 (2): 7-13.
18. Llop P, Latorre A, Moya A. 2018. Experimental Epidemiology of Antibiotic Resistance: Looking for an Appropriate Animal Model System. *Microbiol Spectr* 6.
19. Lo SW, Gladstone RA, Tonder AJ van, Plessis MD, Cornick JE, Hawkins PA, Madhi SA et al. 2019. A mosaic tetracycline resistance gene tet(S/M) detected in an MDR pneumococcal CC230 lineage that underwent capsular switching in South Africa. *J Antimicrob Chemother* 75:512–520.
20. Moullan N, Mouchiroud L, Wang X, Ryu D, Williams EG, Mottis A, Jovaisaite V, Frochoux MV, Quiros PM, Deplancke B, Houtkooper RH, Auwerx J. 2015. Tetracyclines Disturb Mitochondrial Function across Eukaryotic Models: A Call for Caution in Biomedical Research. *Cell Rep* 10:1681–1691.

21. Muñoz-Benavent M, Pérez-Cobas AE, García-Ferris C, Moya A, Latorre A. 2021. Insects' potential: Understanding the functional role of their gut microbiome. *J Pharm Biomed Anal* 194:113787.
22. Onchuru TO, Martinez A, Ingham CS, Kaltenpoth M. 2018. Transmission of mutualistic bacteria in social and gregarious insects. *Curr Opin Insect Sci* 28:50–58.
23. Ourry M, Lopez V, Hervé M, Lebreton L, Mougél C, Outreman Y, Poinso D, Cortesero AM. 2020. Long-lasting effects of antibiotics on bacterial communities of adult flies. *FEMS Microbiol Ecol* 96.
24. Rawat N, Anjali, Shreyata, Sabu B, Jamwal R, Devi PP, Yadav K, Raina HS, Rajagopal R. 2023. Understanding the role of insects in the acquisition and transmission of antibiotic resistance. *Sci Total Environ* 858:159805.
25. Roberts MC. 1990. Characterization of the Tet M determinants in urogenital and respiratory bacteria. *Antimicrob Agents Chemother* 34:476–478.
26. Roberts MC. 2005. Update on acquired tetracycline resistance genes. *FEMS Microbiol Lett* 245:195–203.
27. Sessitsch A, Wakelin S, Schloter M, Maguin E, Cernava T, Champomier-Verges M-C, Charles TC et al. 2023. Microbiome Interconnectedness throughout Environments with Major Consequences for Healthy People and a Healthy Planet. *Microbiol Mol Biol Rev* 87:e00212-22.
28. Sun H, Li H, Zhang X, Liu Y, Chen H, Zheng L, Zhai Y, Zheng H. 2023. The honeybee gut resistome and its role in antibiotic resistance dissemination. *Integrative Zoology* 2023; 0: 1–13.
29. Tinker KA, Ottesen EA. 2020. Phylosymbiosis across Deeply Diverging Lineages of Omnivorous Cockroaches (Order Blattodea). *Appl Environ Microbiol* 86.
30. Vujkovic-Cvijin I, Sklar J, Jiang L, Natarajan L, Knight R, Belkaid Y. 2020. Host variables confound gut microbiota studies of human disease. *Nature* 587:448–454.
31. Zangl L, Kunz G, Berg C, Koblmüller S. 2019. First records of the parthenogenetic Surinam cockroach *Pycnoscelus surinamensis* (Insecta: Blattodea: Blaberidae) for Central Europe. *J Appl Entomol* 143:308–313.
32. Zheng H, Steele MI, Leonard SP, Motta EVS, Moran NA. 2018. Honey bees as models for gut microbiota research. *Lab Anim* 47:317–325.

References not included in the manuscript, but cited in our response to Reviewer #1:

1. Onwugamba FC, Fitzgerald JR, Rochon K, Guardabassi L, Alabi A, Kühne S, Grobusch MP, Schaumburg F. 2018. The role of 'filth flies' in the spread of antimicrobial resistance. *Travel Med Infect Dis* 22:8–17.
2. Fukuda A, Usui M, Okamura M, Dong-Liang H, Tamura Y. 2019. Role of Flies in the Maintenance of Antimicrobial Resistance in Farm Environments. *Microb Drug Resist* 25:127–132).

Re: mSystems01018-23R1 (Transmission of antimicrobial resistance in the gut microbiome of gregarious cockroaches: the importance of interaction between antibiotic exposed and non-exposed populations)

Dear Ms. Amalia Bogri:

Your manuscript has been accepted, and I am forwarding it to the ASM production staff for publication. Your paper will first be checked to make sure all elements meet the technical requirements. ASM staff will contact you if anything needs to be revised before copyediting and production can begin. Otherwise, you will be notified when your proofs are ready to be viewed.

Featured Image Submissions: If you would like to submit a potential Featured Image, please email a file and a short legend to mSystems@asmusa.org. Please note that we can only consider images that (i) the authors created or own and (ii) have not been previously published. By submitting, you agree that the image can be used under the same terms as the published article. File requirements: square dimensions (4" x 4"), 300 dpi resolution, RGB colorspace, TIF file format.

Sincerely,
Jonathan Klassen
Editor
mSystems